# Delivering Musculoskeletal Rehabilitation in the Digital Era: A Perspective on Clinical Strategies for Remote Practice

**DOI:** 10.3390/healthcare13182286

**Published:** 2025-09-12

**Authors:** Muhammad Alrwaily

**Affiliations:** Dr. Alrwaily Academy, Riyadh 13247, Saudi Arabia; muhammad.alrwaily@gmail.com

**Keywords:** telerehabilitation, telehealth, musculoskeletal disorders, musculoskeletal care, virtual care

## Abstract

The purpose of this perspective is to present a structured framework for delivering musculoskeletal (MSK) care via telerehabilitation, advocating for a fundamental shift in the mindset of physical therapists. Rather than viewing virtual care as a limited substitute, it is redefined as a clinically valid model that requires deliberate reengineering of traditional assessment and treatment practices. The article addresses three key questions: (1) How can MSK assessment and treatment be effectively delivered in the digital environment? (2) What clinical reasoning pathways can guide patient triage in virtual care? and (3) What value does telerehabilitation offer to both patients and therapists? The article outlines how MSK sessions can be conducted remotely through a systematic approach to preparation, subjective examination, and physical assessment, each adapted to both the constraints and opportunities of the digital environment. Core elements of in-person care are translated into telehealth-compatible formats, including visual observation, patient-guided special tests, and digitally administered patient-reported outcome measures. It further proposes clinical decision pathways that enable therapists to triage patients into three categories: those fully suitable for telehealth, those requiring hybrid care, and those needing referral. The value proposition of MSK telerehabilitation is discussed from both the patient and therapist perspectives, highlighting enhanced accessibility, efficiency, and patient empowerment. The article contrasts the in-person and telerehabilitation models, underscoring the elevated importance of communication, creativity, resourcefulness, and clinical reasoning in virtual contexts. Beyond current challenges such as regulatory ambiguity, reimbursement variability, and digital inequity, the article explores future directions for MSK care. These include integration of wearable technologies, AI-assisted assessments, and an evolving therapist role as a director of care within a digitally enabled system. Ultimately, this article offers not just a model for virtual MSK sessions, but a vision for sustainable, evidence-informed transformation in rehabilitation delivery.

## 1. Introduction

Musculoskeletal (MSK) conditions are one of the most widespread and disabling health concerns globally, affecting over 1.7 billion people and leading to significant loss in productivity and quality of life [1]. While conditions such as low back pain, osteoarthritis, and neck pain are rarely fatal, they are frequently chronic and highly disabling, responsible for nearly 17% of all years lived with disability [2]. Care for such conditions is challenging due to limited and often inequitable access to timely rehabilitation services [3]. In many regions, patients face long wait times, limited availability of physical therapy, and logistical or financial obstacles to receiving care [4]. These access challenges often delay diagnosis and early intervention, allowing conditions to worsen, which exacerbate chronicity and disability [5]. When rehabilitation is inaccessible, MSK conditions become more difficult and more expensive to treat.

To close this gap, digital models of care, particularly telerehabilitation, offer a promising solution [6]. By removing the need for physical presence, telerehabilitation allows physical therapists to reach patients in remote, underserved, or mobility-limited contexts, expanding the reach of care beyond the walls of the clinic [7]. This shift brings with it new responsibilities for therapists, who must learn to deliver care in an environment where hands-on examination is replaced by visual assessment, verbal instruction, and patient self-management. Therapists engaging in virtual care will increasingly encounter a broader range of conditions, health literacy levels, and cultural expectations than those typically seen in traditional outpatient settings [8]. Telerehabilitation represents not only a response to logistical challenges but a strategic evolution in how and where physical therapy is delivered.

A growing body of evidence supports the use of telerehabilitation as a clinically effective alternative to traditional in-person rehabilitation for MSK conditions [9]. Multiple systematic reviews and meta-analyses have demonstrated that telerehabilitation yields comparable outcomes in pain reduction, functional improvement, and patient satisfaction [9,10,11,12]. The challenge now is not whether telerehabilitation works, but how to ensure it is delivered consistently, safely, and equitably. Real-time video sessions, asynchronous exercise instruction, and self-monitoring tools all contribute to its clinical utility, but must be implemented with attention to infrastructure, training, and patient engagement. Telerehabilitation is not a temporary substitute—it is an evidence-based, sustainable model for MSK care [6,9,13].

The purpose of this paper is to explore the value of telerehabilitation, the evolving MSK patient journey, and the clinical shift required from physical therapists. It addresses three key questions: (1) How can MSK assessment and treatment be effectively delivered in the digital environment? (2) What clinical reasoning pathways can guide patient triage in virtual care? and (3) What value does telerehabilitation offer to both patients and therapists? It offers a structured framework for virtual session delivery, outlines adaptations for remote assessment and treatment, examines key barriers, and highlights future directions. By equipping clinicians with the tools and mindset for digital practice, this paper aims to position them at the forefront of modern MSK rehabilitation.

## 2. Reshaping the Musculoskeletal Care Pathway Through Telerehabilitation

In traditional MSK rehabilitation models, a typical pathway begins with injury or pain, followed by a search for a qualified provider, often requiring a physician referral or lengthy interactions with a clinic’s call center. Appointments may be scheduled days or weeks later, requiring patients to rearrange their lives around travel, parking, waiting, and brief therapist interactions often lasting no more than 30 min. After this session, the patient is sent home with printed or verbal instructions, susceptible to forgetfulness, uncertainty, or poor adherence [14,15]. If symptoms change or questions arise, they must wait until the next session, assuming availability exists, only to repeat the cycle.

This model introduces significant indirect costs [10]: time off work, transportation, caregiver coordination, and missed opportunities for early intervention. It also perpetuates care dropout, particularly when follow-up is difficult or access is inconvenient. The result is a high-friction, low-continuity care experience, especially problematic for chronic MSK conditions requiring sustained engagement [15].

Telerehabilitation reshapes this journey through a digitally enabled, streamlined, and patient-centric model [6]. The process begins with the patient initiating care through a mobile app or web portal without needing to call a clinic or wait for administrative confirmation. Assessment can begin immediately through structured digital intake forms, patient-reported outcomes, and asynchronous video uploads. Initial consultations are conducted via video, where therapists leverage patient history and virtual examination to reach a working diagnosis. With tools like guided self-assessments, remote range-of-motion checks, and pain pattern mapping, the therapist engages the patient as an active participant in the evaluation, marking a notable departure from traditional models where patients have often played a more passive role [16]. In many conventional settings, the emphasis is on therapist-led interventions aimed at symptom resolution, with limited time or structure for shared decision-making or active self-monitoring. Telerehabilitation, by contrast, requires and facilitates greater patient involvement from the outset, encouraging ownership, fostering self-efficacy, and aligning treatment with the patient’s lived environment and functional goals. This shift is especially valuable in chronic MSK care, where sustained engagement and behavioral change are often necessary for long-term success.

Once treatment begins, exercise programs are delivered in interactive video formats, supported by in-app reminders, progress dashboards, and therapist feedback. The patient can ask questions between sessions, upload progress clips, or request adjustments, all without needing another in-person appointment. This continuity of care is particularly valuable when symptoms change, or motivation dips allowing the therapist to intervene early. The patient’s home becomes the treatment setting, and with the right guidance, even simple objects like a towel, chair, or doorframe can be used for strengthening, mobilization, or balance training [6,9].

Most importantly, the telerehabilitation journey is adaptive and continuous, not episodic. It prioritizes patient empowerment, real-time feedback, and clinical responsiveness, making MSK care more sustainable and scalable. For patients who previously dropped out due to access, discomfort, or inconvenience, telerehabilitation offers a more intuitive and supportive pathway to recovery, potentially reducing long-term disability and improving population-level outcomes.

## 3. The Value Proposition of MSK Telerehabilitation

Telerehabilitation is not simply the digitization of traditional MSK care; it is a reconfiguration of the value delivered to both patients and providers. Rather than replicating in-person services through a screen, telerehabilitation redefines how care is accessed, experienced, and sustained. To capture this broader impact, we draw on the value pyramid—a framework that categorizes value into four tiers: functional benefits (efficiency, access), emotional benefits (reassurance, motivation), life-changing effects (empowerment, autonomy), and social impact (equity, public health gains) [17]. This structure allows for a more complete understanding of how telerehabilitation enhances the rehabilitation experience beyond clinical outcomes. When viewed through the value pyramid, telerehabilitation offers a multidimensional value proposition that traditional models often fail to deliver.

For patients (Table 1), telerehabilitation value spans every level of the healthcare value pyramid. At the functional level, it reduces common logistical burdens such as commuting, parking, and wait times; delivering care directly into the home and enabling flexible scheduling. This convenience is particularly meaningful for patients with limited mobility, chronic pain, or caregiving responsibilities.

On the emotional level, increased therapist availability through messaging, video check-ins, and digital progress tracking can reduce anxiety and enhance motivation. Patients who feel seen and supported between sessions often demonstrate better adherence and engagement. While the in-person care offers additional emotional benefits through direct human contact, a trusting therapeutic environment, and the nuanced support that face-to-face interaction enables, combining this with telerehabilitation through a hybrid care model can optimize emotional connection by merging the responsiveness and continuity of digital touchpoints with the depth and reassurance of physical presence.

At the life-changing level, telerehabilitation encourages active self-management, transforming the patient from a passive recipient into an empowered agent of their own recovery. With tools like exercise libraries, real-time feedback, and symptom monitoring, patients develop greater confidence and autonomy in managing their MSK conditions.

Finally, at the level of social impact, telerehabilitation holds the potential to expand access to underserved populations—such as rural residents, women facing mobility restrictions, and individuals in low-resource settings—bridging longstanding gaps in timely rehabilitation access. By delivering functional convenience, emotional support, personal empowerment, and social equity, telerehabilitation enhances the patient experience far beyond what is typically possible in conventional care settings.

For therapists (Table 2), telerehabilitation delivers a similarly layered value. Functionally, it improves clinical efficiency by reducing no-shows, enabling flexible scheduling, and streamlining documentation and follow-up tasks. Tools like automated reminders, integrated outcome tracking, and shared digital exercise plans support a more efficient workflow and broader reach. Documentation, which is often a time-consuming and fragmented process in traditional care, can be simplified through telehealth platforms that offer built-in templates, auto-populated forms, and real-time note-taking features during sessions [9]. On the emotional level, therapists often report reduced physical strain, less burnout, and increased professional satisfaction when their role shifts from manual intervention to education, communication, and strategic oversight. At the life-changing level, telerehabilitation expands the therapist’s role—opening doors to new skills such as digital content creation, remote coaching, and data-driven care planning. These innovations allow clinicians to evolve with the changing demands of healthcare while maintaining clinical impact. At the social level, therapists contribute to more equitable care delivery by reaching populations they could not have otherwise served, aligning their practice with broader public health goals. Telerehabilitation not only enhances clinical productivity but also elevates the therapist’s role as a high-impact, tech-enabled healthcare professional.

## 4. Preparing for MSK Telerehabilitation Session

The steps and decision points involved in delivering MSK care via telerehabilitation are summarized in Figure 1. Successful delivery of MSK telerehabilitation depends not only on clinical knowledge but also on the conditions in which that knowledge is applied. Five core dimensions including technical infrastructure, environmental setup, safety planning, data privacy, and communication, represent a point at which the virtual encounter can either be enabled or undermined. Whether these elements serve as facilitators or barriers depends on how deliberately they are addressed [6,13].

The first and most obvious prerequisite is a stable technological foundation. When patients and therapists connect via high-speed internet on devices equipped with adequate cameras and microphones, the virtual environment can support real-time interaction that approximates in-person care. In contrast, poor bandwidth, frequent connection drops, or the use of small-screen devices such as mobile phones degrade the therapist’s ability to observe, assess, and guide movement. In such cases, the technology itself becomes the barrier to care.

Visual and environmental factors contribute significantly to the quality of assessment and treatment. A well-lit room, a clear visual field, and proper camera positioning allow the therapist to analyze posture and movement with confidence. When these visual conditions are met, the therapist can compensate for the lack of physical palpation through observational precision. But if lighting is poor, the room is cluttered, or the camera view is obstructed, the therapist is left to guess, replacing clinical accuracy with approximation. Thus, the physical setting of the session either supports or undermines the therapist’s clinical judgment.

Safety, which is often assumed in clinic settings, must be actively established in remote environments. A simple pre-session safety check such as verifying floor space, asking about dizziness or fall history, confirming the availability of a nearby phone can allow therapists to prescribe functional movements with confidence. Absent such checks, therapists may withhold certain exercises, limiting the effectiveness of the session. Worse, unrecognized safety hazards may place the patient at risk. In this way, preparedness for safety determines the therapist’s ability to prescribe with clinical integrity.

Data privacy and security, while less visible, shape the patient’s willingness to engage. When therapists use secure, encrypted platforms and explain consent clearly, patients are more likely to trust the process and participate fully. On the other hand, if patients feel unsure about who might be listening, whether their data is protected, or whether they are being recorded, they may hold back key information or decline care entirely. In digital environments, trust must be earned not through presence but through platform and process.

Finally, communication becomes the central clinical tool. In the absence of physical touch, therapists must rely on the clarity of their questions, the specificity of their instructions, and the quality of their feedback. Therapists who listen actively, check for understanding, and speak with intention can guide patients with precision and empathy. But without those skills, even the best clinical plan may be poorly delivered, misinterpreted, or ignored. Here, more than anywhere else, the therapist’s voice becomes their primary instrument of care.

Each readiness dimension represents a fork in the road: one path toward enhanced connection, clinical accuracy, and therapeutic progress; the other, toward confusion, disengagement, and risk. Thoughtful preparation in these areas transforms telerehabilitation from a compromise into a powerful, patient-centered mode of care. These accommodations ensure that telerehabilitation remains inclusive, equitable, and effective across age groups, languages, and cultures.

## 5. Preparing for Patient Encounter in Telerehabilitation

The first virtual encounter between therapist and patient establishes the foundation for all clinical reasoning and therapeutic planning that follows [9,13]. It is during this interaction that the therapist must not only gather clinical history but also construct the digital environment in which care will unfold. In telerehabilitation, the success of the session depends as much on the logistics of setup as it does on diagnostic acumen. Thus, the opening questions must serve a dual purpose: to clarify who is receiving care and to define how care will be delivered.

The session begins, as it would in person, by confirming basic demographics. But in the virtual context, the therapist must also ask: “Are you here for yourself, or are you assisting as a caregiver?” Involving family members or aides is common in pediatric, geriatric, or neurologically impaired populations; and knowing who is present informs how instructions are framed, tasks are delegated, and safety is ensured. When caregivers are present, therapists often provide brief, real-time guidance to help them support the session—such as adjusting camera angles, assisting with movement, or monitoring for safety. While this involvement is usually informal, it effectively enhances care delivery.

Equally critical is identifying the type of device being used: laptop, smartphone, or desktop. Each offers distinct advantages and limitations. A laptop may provide mobility and screen size but lack stable positioning. A desktop offers a wide view but may be fixed and difficult to reposition. A smartphone is portable but often held in the hand, leading to shaky video and limited field of view. By understanding the patient’s technology, the therapist can anticipate what visual angles will be possible, how camera repositioning might be managed, and whether the device will support screen sharing or instructional media.

Next, the therapist must confirm the safety of the physical environment by asking “Are you in a safe space where you can move without obstacles?” and “Do you have any history of falls or conditions that make you feel unsteady?” These questions serve not only as fall-risk screens but also determine whether weight-bearing or balance activities can be introduced in the session. When needed, the therapist may ask the patient to use their device’s camera to visually scan the room, ensuring there are no hazards such as loose rugs, obstructed pathways, or poor lighting. A patient’s ability to safely perform functional movements becomes the threshold for intervention planning.

The next set of questions addresses the spatial limitations of the camera setup. Asking “Can you move close to and far from the camera if needed?” is more than a technical inquiry, it dictates what types of assessments can be conducted. If the patient cannot step back far enough for a full-body view, certain gait or posture evaluations may need to be postponed or adapted. If they cannot position the camera low enough, lower limb mechanics may be obscured. These physical limitations of the frame become clinical boundaries within which the therapist must work.

Finally, the therapist must verify the availability of essential tools and environmental items. For the therapist: a checklist, the exercise platform interface, access to outcome measures, and a backup communication channel (e.g., phone). For the patient: a stable chair without wheels, a wall, a towel or resistance band, and adequate floor space. These are not luxuries, they are substitutes for the treatment table, the modality cart, and the therapist’s hands. Knowing what is available enables the therapist to translate clinic-based protocols into household equivalents with confidence and creativity.

By asking these questions and interpreting the answers through a clinical lens, the therapist does more than collect information, they engineer the therapeutic environment [6,9,13]. In telerehabilitation, this act of intentional setup is not a prelude to care; it is the beginning of care itself. A caregiver or assistant may not only help with communication or positioning but may also serve as an extension of the therapist’s reach; holding the camera to adjust angles during movement assessment, assisting with resisted manual muscle testing, or applying light palpation under guided instruction.

## 6. Subjective Assessment in MSK Telerehabilitation

Once the environment has been secured, the technology confirmed, and roles clarified, the next critical phase in the telerehabilitation encounter is subjective assessment. While the physical distance may be new, the clinical discipline remains the same, and in some respects, it becomes even more essential. In MSK care, where up to 80% of diagnoses can be derived from a thorough patient history, the virtual context elevates this process from routine to foundational [18].

In the absence of hands-on examination, the history becomes the therapist’s most powerful diagnostic tool. Without the tactile feedback of palpation, the controlled resistance of manual muscle testing, or the specificity of joint tests, the therapist must depend on verbal precision and patient narrative. A structured, comprehensive approach ensures that no essential element is missed, and that the clinical reasoning is as robust as it would be in person.

The use of the LMNOPQRST framework offers a practical and evidence-informed scaffold for this purpose [19]:L—Location: Where is the pain or problem?M—Mechanism: How did it start? Was there a specific incident?N—New vs. chronic: Is this a new issue, or has it happened before?O—Onset and duration: When did it start, and how long does it last?P—Palliating/provoking: What makes it better or worse?Q—Quality: How does it feel—sharp, dull, burning, etc.?R—Radiation: Does the discomfort travel to other areas?S—Severity: How intense is it, on a scale from 0 to 10?T—Timing and trends: Is it worse at night, during activity, or with rest?

This structure allows the therapist to capture pain behavior, functional limitations, and symptom patterns in a consistent way that supports hypothesis generation. But its value extends beyond information gathering. When used with empathy, appropriate pacing, and attentive listening, this format becomes a vehicle for building trust. Each question offers an opportunity for the patient to feel heard, respected, and validated; particularly important in the virtual space, where relational warmth must travel through a screen.

The therapist’s tone, posture, and presence, conveyed through eye contact, vocal tone, and active paraphrasing, contribute to the therapeutic alliance just as palpation and nonverbal cues would in person. What may feel like a series of clinical questions is, in reality, the construction of both diagnosis and connection.

In a traditional clinic, a brief history is often augmented quickly by physical testing. In telerehabilitation, the process reverses: the history must carry the weight of early diagnostic reasoning. The therapist must listen for pattern recognition, identify red flags, and begin planning movement screening, all from narrative cues. A well-conducted history becomes not just the first step in care, but the primary anchor for decision-making.

## 7. Integrating Outcome Measures in MSK Telerehabilitation

To support clinical decision-making and monitor patient progress, the use of standardized outcome measures is essential in MSK telerehabilitation. These tools help quantify symptoms, function, and quality of life while offering a baseline for goal setting and longitudinal comparison. In the context of remote care, therapists must be strategic about when and how these tools are administered to ensure relevance without overwhelming the patient. Balancing efficiency with clinical rigor requires that outcome measures be adapted to the telehealth format.

Longer, multidimensional outcome instruments—such as the Oswestry Disability Index [20], Neck Disability Index [21], or the Lower Extremity Functional Scale [22]—are best completed by the patient prior to the initial session. When sent electronically in advance via secure links, patient portals, or mobile apps, these measures allow the therapist to review the results ahead of time, identify red flags, and tailor the subjective assessment accordingly. This pre-session completion also saves valuable time during the video consultation and enhances session efficiency. In contrast, shorter tools—such as the Numeric Pain Rating Scale [23], Global Rating of Change [24], or a brief functional question (“What activities are you currently limited in?”) can be seamlessly incorporated into the subjective examination. These quick assessments fit naturally into the conversational flow of a virtual visit and provide immediate clinical insight.

Moreover, repeated use of these measures throughout the episode of care enables therapists to track improvement, adapt interventions, and justify discharge decisions or referrals. The frequency of re-administration varies based on clinical need: brief tools may be used at each session, while longer forms may be used at intake, mid-point, and discharge. Therapists should also ensure that patients understand the purpose of each tool, correctly interpret them, and how their responses inform treatment. This may involve brief clarifications during the session or providing brief written or video instructions during the pre-session, as misinterpretation can compromise data validity. In a telehealth environment, outcome measures serve not only as clinical data points, but also as vehicles for patient engagement, goal setting, and evidence-informed care.

## 8. Administering Psychosocial Tools in MSK Telerehabilitation

Psychosocial factors such as fear-avoidance, catastrophizing, anxiety, and depression play a critical role in the prognosis and recovery trajectory of musculoskeletal conditions. In the telerehabilitation setting, where body language and subtle emotional cues may be harder to detect, structured psychosocial screening tools offer valuable insight into the patient’s beliefs, attitudes, and readiness for engagement. Integrating these tools early in care can improve triage accuracy, personalize treatment plans, and flag the need for multidisciplinary support. Nevertheless, therapists should ensure that patients understand the intent and content of psychosocial questions. When sending forms electronically, supplemental explanations may be necessary to help the patient interpret the items correctly and respond meaningfully.

Ideally, validated psychosocial instruments such as the Tampa Scale for Kinesiophobia [25], Pain Catastrophizing Scale [26], or the STarT Back Screening Tool [27], should be administered prior to the initial session alongside physical outcome measures. Sending them electronically in advance allows patients to complete them in a private, unrushed environment, which may improve the honesty and accuracy of their responses. This also gives the therapist time to review results and formulate tailored questions for subjective assessment. In cases where advance completion is not feasible, shorter tools or targeted screening questions (e.g., “Do you feel anxious about moving because of your pain?”) can be incorporated into the initial interview.

Therapists should introduce these tools with clarity and empathy, explaining that understanding emotional and behavioral factors helps deliver more effective, whole-person care, not that it implies symptoms are “all in the head.” The administration of such tools should not be limited to the initial session but can be repeated at midpoint or near discharge to track changes in cognitive or emotional barriers, support treatment adjustments, and inform decisions about progression. When indicated, elevated scores should prompt discussion and, if necessary, referral to psychology or pain management. Used skillfully, psychosocial tools in telerehabilitation deepen clinical insight, support biopsychosocial care planning, and build therapeutic alliance in the absence of physical presence.

## 9. Adapting the Physical Examination for MSK Telerehabilitation

In telerehabilitation, the physical examination must be intentionally designed, environmentally adaptive, and diagnostically sound. The therapist no longer operates within a controlled, standardized clinical space. Instead, each patient encounter unfolds in a unique, patient-defined environment shaped by the layout of their home, the type of device used, the presence of a caregiver, and individual safety considerations. For this reason, the therapist must approach each examination not as a replication of the in-clinic model, but as an informed improvisation, built upon prior knowledge gathered from the patient intake and digital readiness checklist. Table 3 shows how some physical examination components can be adapted to virtual environment. Note that these adaptations are informed by clinical experience and have yet to be supported by empirical validation.

Before proceeding with the virtual physical examination, it is essential for the therapist to share an initial impression with the patient describing in clear terms what structures may be involved, such as muscle, ligament, joint, or nerve. This helps frame the purpose of the upcoming virtual examination and sets expectations. The therapist should explain that the next step will involve guided movement testing and self-examination strategies to help clarify the diagnosis. The therapist should emphasize the importance of patient cooperation and feedback, which fosters a collaborative atmosphere, builds therapeutic alliance, and reinforces the patient’s active role in the assessment process.

The physical examination in virtual settings is not inferior to in-clinic assessment; it is simply different [28]. A growing body of evidence supports its validity and reliability, particularly when structured protocols are followed [6,29]. Systematic reviews have demonstrated that virtual evaluations for conditions affecting the shoulder, low back, neck, and knee show moderate to high interrater reliability when conducted via video [28,30,31,32]. Objective assessments including range of motion, strength, and special tests have demonstrated strong concurrent validity and reliability, with intraclass correlation coefficients frequently exceeding 0.90 [6]. Validation studies show that many special tests (e.g., Neer, Hawkins-Kennedy, straight leg raise) can be reliably self-performed or assisted by a caregiver under therapist instruction, yielding diagnostic accuracy comparable to in-person evaluation [33].

Therapists should plan the exam with efficiency and patient energy in mind. Tasks that can be completed in a seated position such as cervical ROM, arm elevation, or neurodynamic testing should be grouped together. Standing, walking, or repeated-movement tasks should be clustered separately. This sequencing strategy, supported by telerehabilitation implementation research, helps minimize cognitive load, fatigue, and unnecessary camera repositioning. It also aligns with best practices for maintaining assessment flow and patient engagement.

Clear and concise communication is central. Without tactile input, the therapist must rely on verbal cues and visual guidance to replace the hands. Instructions should be brief, direct, and reinforced with screen sharing, modeled demonstrations, or embedded instructional videos. Research shows that the clarity of instruction and video quality directly influence diagnostic accuracy in remote MSK exams [9]. Therapists who use pre-recorded demonstrations or structured scripts tend to achieve more consistent assessments. Building rapport is also essential; therapists should occasionally look directly into the camera to simulate eye contact and explain when they need to glance away to take notes. Supplementing the session with high-resolution images, body charts, or diagrams can further support patient understanding and proper self-performance of exam maneuvers.

Even in the absence of hands-on palpation, virtual physical exams can yield meaningful clinical insight. Patients can be asked to point to their area of pain, while the therapist guides with clarifying questions to localize symptoms. When direct palpation is indicated, therapists can instruct patients (or caregivers) to apply gentle pressure to anatomical landmarks and report symptom reproduction. If this is not feasible, the therapist can reference trigger point pain referral maps (e.g., Travell and Simons) [34], screen-share annotated images, and ask the patient whether the pattern matches their own pain experience. Annotated body charts may also be sent in advance to help guide self-palpation. Patients can be taught to locate key bony landmarks such as the posterior superior iliac spine, greater trochanter, or cervical spinous processes using a single finger as described in telehealth protocols.

Following palpation, movement observation becomes a central component of virtual MSK assessment. The therapist evaluates global and regional posture, then guides the patient through active movements to assess coordination, asymmetries, muscle imbalances, atrophy, hypertrophy, and compensatory strategies. To interpret these findings systematically, therapists may utilize frameworks such as Movement System Impairment or Functional Movement Screen [35,36], which evaluate movement direction, regional interdependence, motor control, and symptom behavior rather than isolated anatomical diagnoses. This is especially applicable in virtual care, where compensatory movements may suggest muscle dominance, inhibition, or joint stiffness, instability that would otherwise be identified through hands-on testing. These observations can be captured using screenshots or video segments for later review or progress tracking.

Special tests adapted for self-performance or caregiver assistance continue to offer clinical value. For example, resisted shoulder movements can be performed using towels, walls, or household objects for resistance. Lower quadrant screens can be replicated through clear instructions and visual supervision. Literature shows that many of these adaptations preserve high sensitivity and acceptable specificity, especially when interpreted in the context of a solid clinical hypothesis [9,33]. Mayo Clinic protocols provide practical guidance for adapting tests such as the FABER, Spurling, O’Brien, and Speed’s tests for home execution, using household props and caregiver support when needed [33]. Still, therapists must exercise judgment as most tests were validated in-person, and diagnostic accuracy may vary when performed remotely. In complex cases, such as patients with significant disability, chronic pain, or recent surgery, tasks like self-palpation or functional loading may be impractical. In such cases, a timely in-person evaluation is warranted.

What ultimately unifies this approach is the therapist’s mindset. Rather than viewing the absence of hands-on contact as a deficit, the therapist embraces the variability of the patient’s environment as an opportunity for more functional, real-world assessment. The digital physical exam is not a diminished version of care; it is a reoriented model that shifts the center of gravity from hands to eyes, from contact to connection, from equipment to creativity.

## 10. Triaging in MSK Telerehabilitation

Following assessment and outcome measures, the therapist’s next critical responsibility is to determine the most appropriate clinical pathway. This triaging process is guided by the therapist’s working hypothesis and clinical reasoning, integrating the patient’s history, red flags, symptom patterns, and functional limitations [37]. The accuracy of this step is pivotal not only for safety and effectiveness but also for maintaining trust in remote care models. A clearly communicated clinical direction ensures that patients are routed to the right level of care at the right time.

Based on clinical experience, there are three primary pathways a patient may follow after initial evaluation (Figure 1):

### 10.1. Manageable via Online-Only Care

Patients whose symptoms are mechanical, stable, and not associated with serious pathology often fall into this category. Examples include non-traumatic low back pain, gradual-onset shoulder impingement, or chronic neck pain without neurological signs. These individuals are typically able to perform home exercises, tolerate virtual instruction, and respond well to behaviorally informed education. For this group, online care offers convenience without compromising outcomes.

### 10.2. Online Assessment with In-Person Follow-Up

Some patients may be suitable for initial telehealth screening but present with complexities that require hands-on examination or intervention. These may include suspected radiculopathies, joint instability, or post-operative cases that need manual therapy, specific testing, or modalities not deliverable remotely. In such cases, the therapist should communicate the need for hybrid care and coordinate appropriately. Hybrid models may include in-person follow-up at a healthcare facility or through mobile physical therapy, where the therapist visits the patient at home. Both settings ensure that hands-on assessment and treatment can complement virtual care when needed. This blended model preserves the benefits of telerehabilitation while ensuring clinical completeness.

### 10.3. Referral to Another Healthcare Provider or Service

If red flags are identified such as signs of infection, cancer, fracture, vascular compromise, or systemic disease the patient should not continue along a rehabilitation pathway. Likewise, acute psychiatric symptoms, unexplained weight loss with pain, or new-onset neurological deficits may indicate conditions outside the scope of physiotherapy. These patients should be referred to the appropriate medical specialist, such as a primary care physician, neurologist, or emergency services, depending on urgency. Timely triage and referral reinforce the therapist’s role as a gatekeeper of safe, patient-centered care.

Each of these decisions should be communicated transparently to the patient, with rationale explained in accessible language. By integrating triage with early hypothesis sharing, therapists can guide patients with clarity, build trust, and ensure alignment between care delivery and clinical need. Effective triage is the bridge between virtual assessment and value-based, personalized rehabilitation.

## 11. Delivering Treatment in Telerehabilitation

Once the clinical hypothesis has been established and the care model agreed upon, the treatment phase of telerehabilitation begins; not in a clinic room equipped with modalities and instruments, but in the patient’s living space, using what is available. Here, the therapist’s ability to adapt, educate, and creatively prescribe becomes central. In this setting, clinical expertise is inseparable from clinical inventiveness.

The first therapeutic task is education. The therapist must explain the suspected condition in clear, patient-centered terms. This explanation, whether for rotator cuff tendinopathy, mechanical low back pain, or patellofemoral dysfunction, should outline the nature of the condition, its common causes, expected course, and the rationale behind each proposed intervention. It also should also discuss differential diagnosis, explain what cannot yet be ruled in or out based on the remote exam, and outline the next steps for confirmation (e.g., trial of therapy, referral for imaging, in-person visit if warranted). Importantly, the therapist should invite questions and confirm the patient’s perspective: “Does this explanation make sense based on what you’re feeling?” This interaction not only clarifies the plan but reinforces a sense of therapeutic alliance which is an essential ingredient in virtual care.

Before selecting specific treatment techniques, the therapist must first identify the primary rehabilitation need. Drawing from the Treatment-Based Classification [37,38], this begins with determining whether the patient primarily presents with a mobility deficit or a control deficit. Mobility deficits may involve local impairments in joint capsule flexibility, neural mobility, or muscle extensibility, whereas control deficits reflect issues with neuromuscular activation, motor control acquisition, or movement pattern assimilation. This clinical reasoning step informs not only what to treat, but how to sequence interventions and select appropriate progressions.

In parallel, the therapist should assess where the patient lies along the spectrum of acuity, whether acute, subacute, or chronic, as this influences the intensity, dosing, and goals of care. Patients in the acute phase may require pain modulation, reassurance, and behaviorally informed education, whereas chronic presentations may call for graded exposure and self-efficacy building. Importantly, the therapist must also screen for psychosocial factors that may influence recovery, including fear-avoidance beliefs, low self-efficacy, or signs of emotional distress. When identified, these factors should be acknowledged, addressed through education, and, if needed, supported through referral or behavioral interventions.

The therapist should consider the patient’s functional limitations and how these manifest in their own environment. The inability to reach overhead kitchen shelves, rise from a sofa, or walk the dog can all serve as meaningful anchors for goal setting and home-based intervention planning. Rather than relying solely on impairment-focused metrics, treatment in telerehabilitation can be framed around the restoration of personally relevant, ecologically valid functions that matter to the patient’s daily life.

### 11.1. Interventions for Mobility Deficit

Patients with mobility deficits often present with limitations in joint range, neural mobility, or soft tissue extensibility. These impairments are traditionally addressed in the clinic through manual therapy, soft tissue mobilization, and targeted stretching [38]. In the telerehabilitation setting, these interventions can be repurposed for the virtual setting and adapted for home use with accessible tools and clear instruction.

Joint mobilization techniques can be adapted for the virtual setting. While direct manual mobilizations are not possible remotely, patients can be taught, self-mobilization strategies. For example, belt-assisted lumbar mobilization, towel-based posterior capsule stretches for the shoulder, or using furniture to facilitate anterior hip mobilization. These movements, demonstrated and coached through screen sharing or video guidance, can restore localized joint play and tissue extensibility over time.

Soft tissue mobilization can also be self-applied or performed with the help of a caregiver. Patients can use objects such as tennis balls, foam rollers, rolling pins, or massage sticks to replicate therapist-guided techniques. Instrument-assisted soft tissue mobilization can be mimicked using kitchen utensils like soup spoons or commercially available scraping tools. Trigger point release can be facilitated with lacrosse balls, frozen water bottles, or massage rollers placed against a wall or floor, allowing patients to manage myofascial tightness independently under therapist guidance.

Neural mobility exercises are equally adaptable. Therapists can guide patients through upper and lower limb neurodynamic sequences such as median nerve sliders or sciatic nerve glides, using screen-shared visuals or live demonstration. Real-time observation and cueing ensure correct performance and safe symptom provocation, enabling patients to manage neurodynamic sensitivity from their own environment. Table 4 shows how in-person interventions can be adapted to virtual environment. Note that these adaptations are informed by clinical experience and yet to be validated in the literature.

However, it is important to recognize that certain manual therapy techniques demand precise, skilled hands-on execution that cannot be fully replicated in a virtual setting. Accordingly, certain clinical cases may warrant a hybrid model of care, combining the convenience and continuity of virtual sessions with periodic in-person visits.

### 11.2. Interventions for Control Deficit

Control deficits involve impairments in neuromuscular coordination, motor activation, and movement pattern integration [38]. In the telerehabilitation context, these deficits are addressed through a structured progression that includes neuromuscular activation, motor control acquisition, or movement pattern assimilation (Figure 2).

The first stage focuses on isolated muscle activation and awareness. Patients may perform low-load isometric holds or specific motor recruitment drills using pillows, resistance from walls, or tension from towels to engage target muscles. For example, activating the transversus abdominis or gluteus medius can be achieved with verbal cues and real-time visual feedback using mirrors or the camera feed itself. The therapist’s role is to guide awareness and correct subtle compensations during these foundational drills.

Once isolated control is achieved, patients can progress to the acquisition phase, where they learn to coordinate movement patterns such as squatting, bridging, or scapular control tasks. These exercises are taught using screen-shared instructional videos, followed by therapist demonstration and real-time correction. This stage emphasizes movement precision, tempo, and symmetry, which is critical for building durable motor control.

The final stage is movement assimilation, where control is embedded into functional and meaningful activities. Patients may practice sit-to-stand transitions, step-ups, or gait pattern correction using cues relevant to their home environment. Functional deficits like reaching for high cabinets or lifting objects off the floor become personalized rehabilitation goals. Household items such as water bottles, canned goods, or elastic belts can be used to load these movements without requiring specialized equipment.

For patients with proprioceptive or balance deficits, therapists can integrate balance training using couch cushions, yoga mats, or unstable surfaces available at home. Tasks may be progressed with visual challenges (e.g., eyes closed), cognitive dual-tasks, or dynamic reaching. These exercises not only retrain sensory motor control but also reinforce confidence in functional balance.

Throughout each of these phases, the therapist utilizes the virtual platform to enhance instruction: screen sharing, slow-motion video review, and asynchronous video feedback allow for precise coaching and monitoring. The adaptability of these interventions highlights the therapist’s evolving role, not just as a clinician, but as an innovator designing personalized care in real-world environments.

### 11.3. Continuity of Care and Long-Term Monitoring

To support long-term monitoring and adherence in telerehabilitation, several strategies can be integrated into the digital care model. These include the use of outcome-tracking tools such as patient portals and mobile applications, which allow individuals to regularly log symptoms, pain levels, and activity adherence. Wearable devices can provide objective data on movement, activity, and biometrics, which can be shared with clinicians for ongoing assessment. Automated reminders and digital alerts help maintain exercise compliance, while asynchronous check-ins such as periodic surveys or video updates ensure continued engagement.

Data dashboards within telerehabilitation platforms can visualize patient progress, flag setbacks, and support adaptive care planning. Additionally, gamification strategies, including progress milestones and motivational prompts, may enhance patient engagement and motivation over time. Collectively, these approaches allow telerehabilitation to deliver continuity of care beyond scheduled sessions and support sustainable self-management.

## 12. The Resourceful Therapist

Telerehabilitation challenges the traditional notion that effective care requires specialized equipment or clinical infrastructure. In the remote setting, the therapist’s value is not derived from access to tools, but from their ability to creatively translate clinical intent into the patient’s environment. This shift gives rise to a new archetype in modern MSK care: the resourceful therapist.

The resourceful therapist adapts interventions to what the patient has at home; be it a towel in place of a resistance band, a soup can instead of a dumbbell, or a tennis ball for soft tissue release. Every household object becomes a potential therapeutic tool, and every room in the home becomes a functional training environment. What matters most is not what’s missing from the clinic, but what’s possible with what’s available.

This resourcefulness is not limited to physical substitutions. It extends to how therapists prescribe, deliver, and reinforce care. Instructional videos can be embedded within telerehabilitation platforms or shared securely, giving patients the ability to review techniques between sessions. Screen sharing can be used to demonstrate movements in real time, and video recording tools allow for asynchronous feedback. In doing so, the therapist empowers the patient to take ownership of their progress while ensuring safety and accuracy.

Moreover, therapists often serve as curators of accessible solutions. When patients ask, “What can I use at home?” or “What’s a good substitute at the gym?”, the resourceful therapist provides recommendations that align with the patient’s goals, budget, and physical environment—whether it is a foam roller, a blood flow cuff, or a cable machine alternative.

Ultimately, the resourceful therapist is not just a clinician, but also a teacher, problem-solver, and innovator. Their success in virtual care is defined by their ability to think adaptively, communicate clearly, and translate evidence-based practice into the realities of everyday life. In this way, telerehabilitation does not dilute the quality of care; it amplifies the creativity and intentionality behind it.

## 13. Challenges to the Adoption of MSK Telerehabilitation

While telerehabilitation has shown strong potential in delivering safe, effective, and scalable MSK care, its adoption remains far from universal. The resistance is not due to lack of clinical efficacy but rather to a network of challenges that span health systems, clinicians, and patients alike [8]. Understanding these barriers is not only necessary for implementation, but essential for sustainability [39].

At the system level, perhaps the most pressing challenge is regulatory ambiguity. In many jurisdictions, the legal scope of telehealth practice remains underdefined. Therapists may be unclear on licensing restrictions, reimbursement eligibility, or documentation standards. Without consistent regulatory frameworks, even motivated clinicians may hesitate to offer virtual services, fearing legal repercussions or non-payment. This lack of policy clarity inhibits integration and stalls momentum at the institutional level [40].

Compounding this issue is the fragmented reimbursement landscape. While some payers offer parity between virtual and in-person sessions, others do not. Many private insurers still lack clear billing codes or impose limitations on session frequency, therapist type, or intervention category. For practitioners operating in public systems or under value-based care models, it is often unclear whether outcomes tracked virtually will count toward performance benchmarks. Thus, the economic viability of MSK telerehabilitation remains uncertain in many regions.

However, current trends indicate gradual progress. Several countries have extended temporary telehealth provisions introduced during the pandemic into permanent policy. For example, Australia and parts of the United States have introduced telehealth parity laws and unified billing codes to ensure virtual care is reimbursed comparably to in-person care [41]. Moreover, professional associations such as the American Physical Therapy Association have issued guidance to support therapists in navigating evolving regulations and documentation practices [32].

From the clinician’s perspective, the shift to telehealth demands a fundamental recalibration of professional identity and skillset as this evolution in care delivery also comes with a cost. While telerehabilitation enhances the therapist’s role as a high-impact, tech-enabled clinician, it simultaneously introduces a workload that must be acknowledged and supported for long-term sustainability. Therapists trained in hands-on assessment and manual intervention may feel unprepared—or philosophically resistant—to delivering care through a screen. Adapting physical examination procedures, building rapport without touch, and demonstrating exercises without tactile correction all require new competencies that are often absent from traditional physical therapy curricula. Without structured training, even experienced therapists may doubt their own effectiveness in a virtual model.

To overcome these barriers, academic institutions should incorporate telerehabilitation modules into entry-level curricula and continuing education, including virtual case simulations, remote communication strategies, and technology onboarding. Moreover, experienced clinicians can serve as digital mentors, modeling confidence and success in virtual care delivery.

On the patient side, digital literacy, technological access, and environmental readiness pose real obstacles [8]. Older adults, those from lower socioeconomic backgrounds, or individuals with cognitive impairments may struggle to navigate platforms, position their cameras, or follow digital instructions [42]. Even when technology is available, the physical environment may not support movement-based assessment; rooms may be too small, lighting too poor, or privacy too limited. These barriers do not reflect a lack of need, but a mismatch between platform design and patient reality.

To overcome these disparities, several practical solutions can enhance accessibility. Providing simplified interfaces, multilingual platform options, and step-by-step onboarding guides can help reduce cognitive and technical demands. Short pre-session orientation calls and the involvement of caregivers or family members during setup may support digital readiness for older adults. The United Kingdom, for example, has developed national digital MSK pathways through initiatives like the National Health Service MSK Digital Playbook [43], integrating virtual assessment and remote management into care delivery. These models streamline triage, improve access, and support continuity across services. They serve as practical blueprints for scalable telerehabilitation.

Finally, a more subtle but equally powerful barrier is cultural expectation. Patients accustomed to in-person care may equate physical presence with quality. Without visual cues like clinic branding, uniforms, or professional equipment, some may perceive telehealth as less legitimate even if the outcomes are equal. This perception problem is particularly challenging in MSK care, where manual therapy has been traditionally overemphasized in both clinical marketing and patient expectation.

Addressing these challenges requires system-level planning, clinician retraining, and patient education. No single solution will suffice. Policies must be modernized, digital platforms optimized, and therapists re-equipped; not only with technical tools, but with confidence in the value of remote care. If these barriers are left unaddressed, telerehabilitation risks remaining a niche offering rather than a core component of MSK healthcare delivery.

## 14. Swot Analysis of MSK Telerehabilitation

Given the multilayered nature of telerehabilitation implementation, a structured analysis can help clarify where the model thrives and where further development is required [44,45]. Drawing on selected insights from this paper, the following SWOT analysis (Table 5) highlights key strengths, weaknesses, opportunities, and threats related to the delivery of musculoskeletal telerehabilitation. This format not only aids critical evaluation but also supports strategic decision-making for clinicians, educators, and policymakers.

## 15. The Future of Telerehabilitation

As telerehabilitation continues its evolution, its future lies not in mimicking traditional clinical care, but in fundamentally reengineering how MSK rehabilitation is assessed, delivered, and experienced. The convergence of digital tools, sensor technology, and artificial intelligence offers the opportunity to move beyond synchronous video encounters into a model of care that is continuous, personalized, and data-driven.

The assessment process, for example, will increasingly leverage capabilities that extend far beyond what a single webcam can offer. Already, clinicians are capturing screenshots and video clips during sessions to document posture, movement patterns, and exercise technique [46]. These visual records enable therapists to establish baselines, track progress over time, and involve patients more actively in recognizing change.

More advanced tools, such as wearable sensors, will allow for real-time tracking of joint angles, gait symmetry, step counts, movement velocity, and load distribution. These wearables, whether integrated into clothing, attached to joints, or worn on the wrist or waist, provide therapists with quantifiable insights into motor performance and adherence. Combined with mobile apps, they can transmit data asynchronously to the therapist, allowing for remote monitoring, early detection of compensation patterns, and objective measurement of outcomes.

Perhaps the most disruptive enabler, however, is artificial intelligence (AI). As machine learning models become capable of interpreting large volumes of patient-generated data therapists will be equipped with predictive tools that support diagnostic reasoning, exercise progression, and risk stratification. Far from replacing clinical judgment, AI will serve as a clinical co-pilot, flagging deviations from expected recovery trajectories and enabling early, data-informed intervention.

With this technological foundation, the treatment experience itself will also transform. As therapists prescribe exercise programs, they will increasingly deliver them through platforms that offer video demonstrations, real-time form correction Via computer vision, and biofeedback from wearables. Sessions will no longer be confined to scheduled calls, but extended across the week through asynchronous feedback, reminders, and adaptive program modifications based on sensor data.

These shifts demand a new professional identity. The physical therapist will no longer function solely as a service provider executing care within the constraints of a 30 min room-based session. Instead, the therapist becomes a *director of care*: an orchestrator of interventions, data interpreter, educator, and strategist. The hands may be distant, but the influence is continuous.

Therapists will advise patients not only on movement and recovery, but also on which wearable technologies to use, which apps to trust, what home tools to buy, and how to track progress meaningfully. They will integrate insights from devices, behavior platforms, and clinical reasoning into a single care plan centered on the patient but informed by a network of digital supports.

In this emerging model, the value of care is no longer tied to physical proximity, but to personalization, continuity, and empowerment. And the therapist, now digitally augmented and strategically positioned, is uniquely equipped to lead this transformation.

## 16. Summary

This perspective paper explores how telerehabilitation is transforming the delivery of MSK care by shifting it from a clinic-dependent, hands-on model to a digitally enabled, patient-centered system. Grounded in evidence and real-world practice, the paper outlines the essential components of effective virtual MSK rehabilitation, from patient readiness and structured history taking to adaptive physical examination and creative treatment strategies.

As shown in Figure 1, a structured virtual encounter begins with environmental and technological preparedness, followed by a comprehensive history guided by frameworks such as LMNOPQRST. Physical examination, while adapted for the digital setting, retains clinical validity through observation, guided self-palpation, and modified special tests. Therapists must sequence movements logically, communicate with clarity, and interpret findings with transparency, inform patients of the working diagnosis, discuss the limitations of virtual assessment, and determine the appropriate care model: fully remote, hybrid, or in-person.

Treatment delivery in telerehabilitation demands innovation. Manual therapy is reimagined as self-mobilization or assisted care, with common household items replacing clinical tools. Wearables, instructional videos, and app-based programs enable continuous engagement, while therapists act not just as providers, but as directors of care guiding patients through recovery across space and time.

Despite structural, educational, and cultural barriers, patient willingness to engage and pay for accessible, effective care is growing. Telerehabilitation offers value across all levels of the healthcare value pyramid: it saves time, reduces stress, empowers patients, and expands access to underserved populations.

It is important to emphasize that telerehabilitation is not intended to replace the role of the human therapist, but rather to extend their reach and enhance care delivery through digital means. Looking forward, the field will be shaped by AI-powered assessment, wearable sensor integration, and the expansion of asynchronous, data-informed care. The physical therapist’s role will continue to evolve, less defined by manual contact and more by clinical orchestration, digital fluency, and strategic decision-making.

Telerehabilitation is not a lesser alternative to in-person care. It is a redefinition of care itself, more personalized, more accessible, and more aligned with the realities of modern life. As technology continues to advance, the opportunity is clear: we can make MSK rehabilitation smarter, more inclusive, and profoundly more human.

## Figures and Tables

**Figure 1 healthcare-13-02286-f001:**
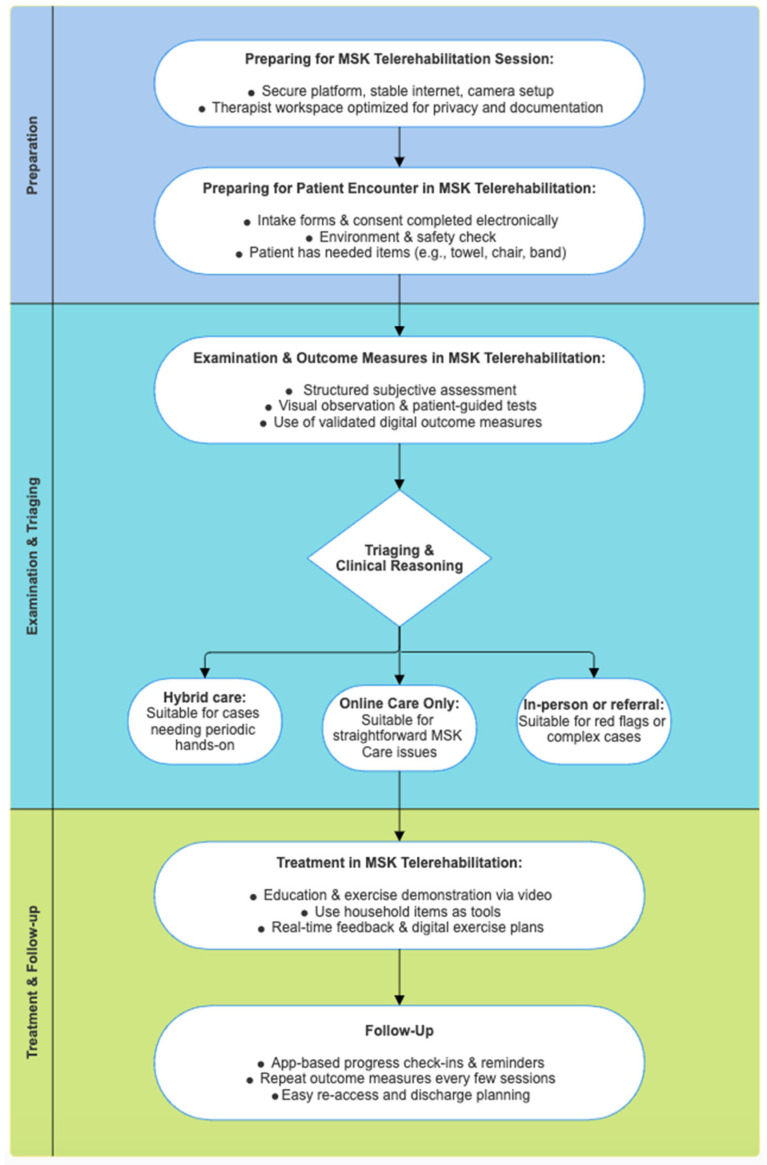
MSK telerehabilitation framework. MSK: Musculoskeletal.

**Figure 2 healthcare-13-02286-f002:**
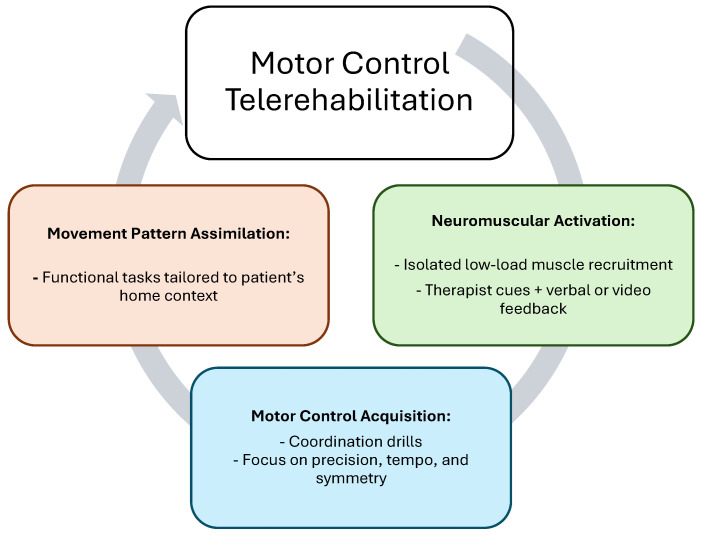
Interventions for control deficits.

**Table 1 healthcare-13-02286-t001:** Telerehabilitation value proposition for patients.

Pyramid Level	Value Elements	How Telerehabilitation Delivers It?
Functional	Saves time, reduces effort, simplifies access	Eliminates travel, parking, waiting; enables at-home care on-demand
Emotional	Reduces anxiety, improves motivation	More frequent communication and reassurance; visual tracking of progress
Life-Changing	Empowers self-care, builds confidence	Patient becomes an active agent in rehab, with on-demand guidance and digital tools
Social Impact	Increases access to care, especially in underserved regions	Bridges rural-urban gaps and reaches those with physical, economic, or social barriers

**Table 2 healthcare-13-02286-t002:** Telerehabilitation value proposition for therapists.

Pyramid Level	Value Elements	How Telerehabilitation Delivers It?
Functional	Increases reach, improves efficiency, supports documentation	See more patients flexibly, reduce no-shows, automate exercise programs and follow-ups
Emotional	Reduces burnout, increases professional satisfaction	Less physical strain; more focus on education, communication, and outcomes
Life-Changing	Enables innovation and skill development	Expands therapist role into coaching, digital content creation, and data-driven practice
Social Impact	Contributes to equitable healthcare delivery	Reaches previously underserved populations and elevates public health through scalable models

**Table 3 healthcare-13-02286-t003:** Adaptation of traditional in-person examination to telerehabilitation.

Traditional Component	Telerehabilitation Adaptation
Controlled clinical environment	Adaptation to patient’s home environment (space setup, lighting, caregiver assistance)
Tactile input/manual contact	Verbal instructions and visual cues (model demonstrations, screen-shared videos/images)
Palpation by therapist	Guided self-palpation, caregiver-assisted palpation, or use of body charts and pain maps
Postural and movement observation	Video-based observation with screen capture, patient-performed functional tasks
Special tests performed by clinician	Self-performed tests using household objects; caregiver-assisted maneuvers when safe
Goniometric measurement	Visual estimation via camera, app-based ROM tracking, or smart device motion sensors
Strength testing (manual resistance)	Functional proxies (e.g., sit-to-stand, single leg balance), self-resistance, or props
Gait assessment	In-place walking observed via full-body camera view or wearable gait sensors
Outcome measures (paper forms)	Digitally administered PROMs (e.g., via app, secure link, or screen share during session)
Real-time documentation	Live note-taking, digital charting, screenshots, and patient-recorded clips
Clinical hypothesis refinement	Enhanced reliance on patient history, verbal symptom description, and movement analysis

PROMs = Patient reported outcome measures.

**Table 4 healthcare-13-02286-t004:** Adaptation of traditional in-person treatment tools to telerehabilitation.

Purpose/Use	Clinic-Based Tool	Adaptation for Telerehabilitation
Soft tissue mobilization	Foam roller	Rolling pin, plastic water bottle, wrapped towel
Trigger point release, myofascial work	Massage ball/Therapy ball	Tennis ball, lacrosse ball, frozen water bottle
Instrument-assisted soft tissue mobilization	Graston tools/IASTM instruments	Stainless steel kitchen utensils, coins, butter knife
Resistance training, stretching	Resistance bands (TheraBand)	Towel, elastic belt, yoga strap, bicycle inner tube
Resistance training	Dumbbells	Water bottles, canned goods, filled grocery bags
Core activation, postural control	Stability ball	Pillow, couch cushion, tightly packed backpack
Joint mobilization	Manual therapy (joint mobilization)	Belt/towel-assisted self-mobilization, wall or chair leverage
Myofascial stimulation, pain modulation	Dry needling	Percussion massage gun, acupressure mat, pointed massage stick
Hypothetical Blood flow restriction (BFR) training	BFR cuffs	Resistance band, compression sleeve (used with caution)
Balance and proprioception training	Balance board/Bosu ball	Couch cushion, folded yoga mat, towel on slippery floor
Hand and finger strengthening	Therapy putty/grip tools	Towel twisting, sponge squeezing, thick rubber bands
Deep neck flexor training	Cervical retraction tools	Wall, resistance from hand or towel
Pain management	TENS/NMES unit	Commercially available TENS/NMES unit
Functional step training, eccentric loading	Step box/Plyo box	Stair steps, low stool, or firm stack of books
Movement quality feedback	Mirrors for feedback	Phone/computer camera, handheld mirror
Patient education and anatomical reference	Clinical posters/models	Screen-shared images, 3D anatomy apps, shared PDFs

IASTM = Instrument-assisted soft tissue mobilization. NMES = Neuromuscular electrical stimulation. TENS = Transcutaneous electrical stimulation.

**Table 5 healthcare-13-02286-t005:** SWOT analysis of MSK telerehabilitation.

**Strengths**	**Weaknesses**
Improved accessibility for patients in rural or underserved areas Flexibility in scheduling and follow-up Efficient use of validated outcome and psychosocial tools Opportunities for enhanced documentation and progress tracking Promotes patient autonomy and engagement through self-management	Lack of tactile/manual assessment tools Variable digital literacy among patients and clinicians Dependence on patient-reported information and self-performance Training gaps in PT curriculum regarding virtual assessment and treatment Environmental constraints in the patient’s home may limit exam or exercise options
**Opportunities**	**Threats**
Expansion of hybrid care models including mobile physio and home visits Integration of wearable tech, AI, and asynchronous monitoring Cross-border consultations and collaborations in global MSK care Curriculum reform to prepare future therapists for digital health	Regulatory ambiguity and inconsistent reimbursement frameworks Skepticism among therapists and patients regarding the legitimacy of virtual care Risk of perceived impersonality or depersonalization of care Technology failures or privacy concerns undermining trust in virtual platforms

AI: Artificial Intelligence; MSK: Musculoskeletal; SWOT: Strengths, Weaknesses, Opportunities, Threats.

## Data Availability

Data sharing is not applicable.

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
