# Peer review of "Delivering Musculoskeletal Rehabilitation in the Digital Era: A Perspective on Clinical Strategies for Remote Practice"

_healthcare, 2025, doi:10.3390/healthcare13182286_

Round 1
Reviewer 1 Report
Comments and Suggestions for Authors
Thank you for the opportunity to review this interesting work that offers an informed, critical, and thoughtful opinion on Musculoskeletal Telerehabilitation.
Below I have listed aspects for improvement:
TITLE:
-Add the type of study presented in the title.
-Complete the author's affiliation information.
KEYWORDS:
-It is recommended to use terms from the MESH thesaurus. The term "Virtual Health" could be replaced with terms such as "Telehealth" or "Telecare."
INTRODUCTION
-Place a period after the citation in the phrase: "timely rehabilitation services."[3]
-Tables 1 and 2 should be resized to the same format and font size. Improve the format of Table 4.
-Provide literature that supports this statement: "Telerehabilitation is not a temporary substitute—it is evidence-based."
-It would be interesting to add scientific literature support in sections 2, 4, 5, and 9.
- THE VALUE PROPOSITION OF MSK TELEREHABILITATION:
I somewhat disagree with the statements about the patient:
-On an emotional level, it is true that greater therapist availability and digital progress monitoring can improve patient motivation. However, the in-person modality offers additional emotional benefits, especially through direct human contact, a trusting environment, and comprehensive support https://doi.org/10.3390/ijerph20054358, https://doi.org/10.1093/ptj/pzab075.
-On the other hand, the social impact is significant: "telerehabilitation expands access to underserved populations..." It should be noted that this is the case as long as digital resources are accessible to this population and they know how to use them.
-For therapists: It is clear that telerehabilitation represents a shift in the care model and an opportunity for change and innovation. However, at some point it should be recognized that generating all the necessary materials for the intervention (documents, forms, videos, training packages, follow-up strategies, etc.) customized for each patient represents a huge amount of work that also requires a significant amount of preparation time.
7 and 8. INTEGRATING OUTCOME MEASURES AND PSYCHOSOCIAL TOOLS IN MSK TELEREHABILITATION
I think it's a great idea for the patient to complete the outcome instruments (Oswestry Disability Index, Neck Disability Index, etc.) and psychosocial tools (Tampa Scale for Kinesiophobia, Pain Catastrophizing Scale, etc.) before the initial session. However, how can the therapist ensure that the patient correctly interprets the questions on these scales and questionnaires? It is recommended that this be indicated.
- I believe that section 9. TRIAGING IN MSK TELEREHABILITATION should be included after section 10. ADAPTING THE PHYSICAL EXAMINATION FOR MSK TELEREHABILITATION.
- There are certain aspects addressed in section 11. DELIVERING TREATMENT IN TELEREHABILITATION, which, given the complexity and the need for precision and skill in the execution of certain manual treatments such as joint, neural, and soft tissue mobilizations, make the professional's hand irreplaceable. Therefore, it is suggested to approach this content with caution.
- On the other hand, theorizing about therapeutic interventions in physiotherapy may be easy; however, putting it into practice successfully and with good therapeutic results is complicated.
AUTHOR CONTRIBUTIONS: Please complete this section correctly.
REFERENCES:
-Review how to prepare the bibliography in the journal's guidelines: https://www.mdpi.com/authors/references
-Include the DOI whenever available.
-When there are more than 6 authors, add et al. after the 6th author.
-Journal titles should be abbreviated.
-Reference No. 1 is not finalized.
Author Response
Reviewer 1
Comment 1: Thank you for the opportunity to review this interesting work that offers an informed, critical, and thoughtful opinion on Musculoskeletal Telerehabilitation.
Response 1: You are welcome. I appreciate your time and comments. Below I will address your valuable comments one-by-one.
Comment 2:
-Add the type of study presented in the title.
-Complete the author's affiliation information.
Response 2: Thank you for pointing this out. I have added the word “perspective” in the title as follows: (Delivering Musculoskeletal Rehabilitation in the Digital Era: A Perspective on Clinical Strategies for Remote Practice). Please see the change in the title highlighted in red.
I also double-checked the name and affiliations, and they are accurate. I also pointed out this in the comment box that is provided.
Comment 3: It is recommended to use terms from the MESH thesaurus. The term "Virtual Health" could be replaced with terms such as "Telehealth" or "Telecare."
Response 3: I have changed the term “virtual health” to the term “Telehealth”. Please find this in the keywords section highlighted in red.
Comment 4: -Place a period after the citation in the phrase: "timely rehabilitation services."[3]
Response 4: Thanks for catching this. I placed the period after citation [3] in the introduction. It is highlighted in red.
Comment 5: -Tables 1 and 2 should be resized to the same format and font size. Improve the format of Table 4.
Response 5: Table 1 and 2 have been resized. Table 4 format has been improved.
Comment 6: Provide literature that supports this statement: "Telerehabilitation is not a temporary substitute—it is evidence-based."
Response 6: I added reference [6,9,13] to the statement in the introduction. I highlighted this change in the introduction section in red.
Comment 7: -It would be interesting to add scientific literature support in sections 2, 4, 5, and 9.
Response 7: references were added to 2, 4, 5, and 9 sections. They are also highlighted in red. I also indicated when the statement is based on clinical experience.
Comment 8: I somewhat disagree with the statements about the patient:
-On an emotional level, it is true that greater therapist availability and digital progress monitoring can improve patient motivation. However, the in-person modality offers additional emotional benefits, especially through direct human contact, a trusting environment, and comprehensive support
Response 8: Thank you for your note. I have included the statement in Section 3 to say:
“On the emotional level, increased therapist availability through messaging, video check-ins, and digital progress tracking can reduce anxiety and enhance motivation. Patients who feel seen and supported between sessions often demonstrate better adherence and engagement. While the in-person care offers additional emotional benefits through direct human contact, a trusting therapeutic environment, and the nuanced support that face-to-face interaction enables, combining this with telerehabilitation through a hybrid care model can optimize emotional connection via merging the responsiveness and continuity of digital touchpoints with the depth and reassurance of physical presence.”
Comment 9: -On the other hand, the social impact is significant: "telerehabilitation expands access to underserved populations..." It should be noted that this is the case as long as digital resources are accessible to this population and they know how to use them.
Response 9: Thank you for your note. I have included the statement in Section 13 to say:
“On the patient side, digital literacy, technological access, and environmental readiness pose real obstacles [35]. Older adults, those from lower socioeconomic backgrounds, or individuals with cognitive impairments may struggle to navigate platforms, position their cameras, or follow digital instructions.”
I discussed this challenge in Section 13 to avoid redundancy in the text
Comment 10: -For therapists: It is clear that telerehabilitation represents a shift in the care model and an opportunity for change and innovation. However, at some point it should be recognized that generating all the necessary materials for the intervention (documents, forms, videos, training packages, follow-up strategies, etc.) customized for each patient represents a huge amount of work that also requires a significant amount of preparation time.
Response 10: Thank you for your note: I have included the statement in Section 13 to say:
“…as this evolution in care delivery also comes with a cost. While telerehabilitation enhances the therapist’s role as a high-impact, tech-enabled clinician, it simultaneously introduces a workload that must be acknowledged and supported for long-term sustainability. Therapists trained in hands-on assessment and manual intervention may feel unprepared—or philosophically resistant—to delivering care through a screen. Adapting physical examination procedures, building rapport without touch, and demonstrating exercises without tactile correction all require new competencies that are often absent from traditional physical therapy curricula. Without structured training, even experienced therapists may doubt their own effectiveness in a virtual model.”
I discussed this challenge in Section 13 to avoid redundancy in the text
Comment 11: I think it's a great idea for the patient to complete the outcome instruments (Oswestry Disability Index, Neck Disability Index, etc.) and psychosocial tools (Tampa Scale for Kinesiophobia, Pain Catastrophizing Scale, etc.) before the initial session. However, how can the therapist ensure that the patient correctly interprets the questions on these scales and questionnaires? It is recommended that this be indicated.
Response 11: I have included a statement in Section 7 that says:
“Therapists must ensure patients correctly interpret outcome tool questions. This may involve brief clarifications during the session or providing brief written or video instructions, as misinterpretation can compromise data validity.”.
I also have included a statement in Section 8 that says:
Nevertheless, therapists should ensure that patients understand the intent and content of psychosocial questions. When sending forms electronically, supplemental explanations may be necessary to help the patient interpret the items correctly and respond meaningfully.
Comment 12: I believe that section 9. TRIAGING IN MSK TELEREHABILITATION should be included after section 10. ADAPTING THE PHYSICAL EXAMINATION FOR MSK TELEREHABILITATION.
Response 12: I have rearranged the sections. Now Section 9 is section 10; and section 10 is section 9.
Comment 13: There are certain aspects addressed in section 11. DELIVERING TREATMENT IN TELEREHABILITATION, which, given the complexity and the need for precision and skill in the execution of certain manual treatments such as joint, neural, and soft tissue mobilizations, make the professional's hand irreplaceable. Therefore, it is suggested to approach this content with caution.
- On the other hand, theorizing about therapeutic interventions in physiotherapy may be easy; however, putting it into practice successfully and with good therapeutic results is complicated.
Response 13: I have included a statement in Section 11.1 that says: “
However, it is important to recognize that certain manual therapy techniques demand precise, skilled hands-on execution that cannot be fully replicated in a virtual setting. Accordingly, certain clinical cases may warrant a hybrid model of care, combining the convenience and continuity of virtual sessions with periodic in-person visits.”
Comment 14: AUTHOR CONTRIBUTIONS: Please complete this section correctly.
Response 14: Author contributions has been completed.
Comment 15: REFERENCES:
-Review how to prepare the bibliography in the journal's guidelines: https://www.mdpi.com/authors/references
-Include the DOI whenever available.
-When there are more than 6 authors, add et al. after the 6th author.
-Journal titles should be abbreviated.
-Reference No. 1 is not finalized.
Response 15: I did review the bibliography as best as I could. I downloaded the mdpi reference style from Endnote website and used it to index my references. Reference 1 is now corrected.
Reviewer 2 Report
Comments and Suggestions for Authors
Dear Autor,
The paper is extremely interesting, providing insight into a field still far from the traditional approach to physical therapy and care provision. This paper had several objectives to present, which were also achieved. The paper offers all the essential components of telerehabilitation that must be satisfied for the process to be realized. I suggest several recommendations that could help make the paper appear more comprehensive.
Author List and Affiliations: Please review the guidelines regarding the listing of authors and their affiliations. Also, the corresponding author is missing.
Abstract:
- There is a blank line visible in the abstract. Please connect the paragraphs of the abstract.
- I suggest clearly highlighting the purpose and objectives of the paper in the abstract. Provide an explicit statement of the main objective(s) or question(s) the manuscript addresses.
- I suggest adding keywords such as MSK Care or Virtual Care or Musculoskeletal Conditions.
- INTRODUCTION:
- I suggest clearly highlighting the purpose and objectives of the paper in the introduction. Provide an explicit statement of the main objective(s) or question(s) the manuscript addresses.
- In the introduction, the sentence: “Care for such conditions is challenging due to limited and often inequitable access to timely rehabilitation services.[3]” – is incorrectly cited – the punctuation mark should be after the reference number.
- RESHAPING THE MUSCULOSKELETAL CARE PATHWAY THROUGH TELEREHABILITATION
- The chapter includes the sentence: “…the therapist engages the patient as an active participant the evaluation.” It would be good to find or comment on data regarding how often patients are passive participants – how much the focus is on quickly resolving the problem rather than on the patient’s active involvement in their own treatment process.
- THE VALUE PROPOSITION OF MSK TELEREHABILITATION
- I suggest standardizing the table presentations. Table 1 is larger than Table 2.
- Regarding Table 2 (Telerehabilitation Value Proposition for Therapist): “Functionally, it improves clinical efficiency by reducing no-shows, enabling flexible scheduling, and streamlining documentation and follow-up tasks” – it would be good to explain how telerehabilitation simplifies or optimizes documentation, which therapists and healthcare staff are otherwise burdened with. Perhaps support with a reference or provide a personal suggestion.
- PREPARING FOR MSK TELEREHABILITATION SESSION
- This chapter lists several dimensions that must be supported to enable telerehabilitation. It raises the question of whether this is feasible with patients of different age groups, cultures, and languages. It might be desirable to provide an explanation or the authors’ personal suggestion on how sessions could be conducted in such cases, emphasizing technical support for telerehabilitation in older adults and communication – multilingualism in patients from different countries.
- PREPARING FOR PATIENT ENCOUNTER IN TELEREHABILITATION
- The role of the family during the telerehabilitation process is emphasized. It would be good to clarify whether the family also undergoes some form of education to assist during the telerehabilitation process.
- It states: “…the therapist must confirm the safety of the physical environment” – how objective is it to rely on the patient’s subjective assessment of the space? It might be necessary to clarify whether it is essential that the therapist conducting telerehabilitation visits the patient’s environment, space, and infrastructure in person for a one-day inspection before starting telerehabilitation.
- SUBJECTIVE ASSESSMENT IN MSK TELEREHABILITATION
- The section emphasizes: “…the therapist must depend on verbal precision and patient narrative” – this raises the question of objectivity in the telerehabilitation process – can we fully rely on the patient’s words even if this means building mutual trust? How can we ensure the reliability of the LMNOPQRST framework?
- INTEGRATING OUTCOME MEASURES IN MSK TELEREHABILITATION
- Please explain in more detail how questionnaires and assessment tools will be sent to patients, how they will complete them, and where they will be stored. What will be the frequency of using such tools given that the basic therapist tools, such as palpation and manual measurements, are absent in the telerehabilitation process – please explain the statement “…repeated use of these measures throughout the episode of care…”
- ADMINISTERING PSYCHOSOCIAL TOOLS IN MSK TELEREHABILITATION
- It is stated: “Integrating these tools early in care can improve triage accuracy, personalize treatment plans, and flag the need for multidisciplinary support.” – it would be desirable to explain in more detail how psychosocial tools could be implemented throughout the telerehabilitation process and what that would mean for the patient.
- TRIAGING IN MSK TELEREHABILITATION
- In the subsection Online Assessment with In-Person Follow Up, it is stated: “In such cases, the therapist should communicate the need for hybrid care and coordinate appropriately.” – it would be good to explain how the hybrid model would be conducted – would the patient come to the healthcare facility or would the therapist visit the patient? There are many therapists in the private sector who conduct mobile physiotherapy – Physio to Go. Could this form of therapy be implemented in a hybrid telerehabilitation model?
- ADAPTING THE PHYSICAL EXAMINATION FOR MSK TELEREHABILITATION
- Regarding the sentence: “Research shows that the clarity of instruction and video quality directly influence diagnostic accuracy in remote MSK exams” – which research/studies are referred to? Please add references.
- Regarding the sentence: “Literature shows that many of these adaptations preserve high sensitivity and acceptable specificity, especially when interpreted in the context of a solid clinical hypothesis” – which research/studies are referred to? Please add references.
- DELIVERING TREATMENT IN TELEREHABILITATION
- I suggest revising Table 4. Place Purpose/Use in the first column, Clinic-Based Tool in the second, and Adaptation for Telerehabilitation in the third. Explain all abbreviations found in the table and include them in the legend.
- Add references for the explanations of Mobility Deficit and Control Deficit.
- I suggest a visual presentation of the three phases of control deficit interventions (Isolated muscle activation, acquisition phase, movement assimilation) and link it to the following chapter on resources (12. THE RESOURCEFUL THERAPIST).
- THE RESOURCEFUL THERAPIST
- I suggest linking this chapter with chapter 11. DELIVERING TREATMENT IN TELEREHABILITATION.
- CHALLENGES TO THE ADOPTION OF MSK TELEHEALTH
- This section lists numerous barriers and challenges faced by telerehabilitation. The biggest problem is therapist motivation and the traditional physical therapy curriculum. It would be excellent to add potential solutions and/or reflections on how to overcome such barriers that are key for the active implementation of telerehabilitation. It would be highly valuable to include examples from different countries that already actively implement telerehabilitation.
- SUMMARY
- Instead of “Summary,” it should be “Conclusion.”
Finally, given the complexity and multilayered nature of the approach considered here, we believe that a SWOT analysis based on selected chapters could serve as a useful framework for discussion and critical evaluation. By including such a structured approach — which has already been used in the analysis of various clinical and educational strategies (e.g., in telemedicine*) — it is possible to further clarify the potentials for the broader application of telerehabilitation in practice. It would be important to emphasize in the conclusion that telerehabilitation does not aim to replace the human being.
- * Caponnetto V, Ornello R, De Matteis E, Papavero SC, Fracasso A, Di Vito G, Lancia L, Ferrara FM, Sacco S. The COVID-19 Pandemic as an Opportunity to Improve Health Care Through a Nurse-Coordinated Multidisciplinary Model in a Headache Specialist Center: The Implementation of a Telemedicine Protocol. Telemed J E Health. 2022 Jul;28(7):1016-1022.
- * Pinto M, Gimigliano F, De Simone S, Costa M, Bianchi AAM, Iolascon G. Post-Acute COVID-19 Rehabilitation Network Proposal: From Intensive to Extensive and Home-Based IT Supported Services. International Journal of Environmental Research and Public Health. 2020; 17(24):9335.
Author Response
Reviewer 2
Comment1: Dear Autor,
The paper is extremely interesting, providing insight into a field still far from the traditional approach to physical therapy and care provision. This paper had several objectives to present, which were also achieved. The paper offers all the essential components of telerehabilitation that must be satisfied for the process to be realized. I suggest several recommendations that could help make the paper appear more comprehensive.
Response 1: Thank you for your comments. I will address your valuable comments one-by-one.
Comment 2: Author List and Affiliations: Please review the guidelines regarding the listing of authors and their affiliations. Also, the corresponding author is missing.
Response2: I have reviewed the Author list and Affiliations. The only author of this manuscript is Muhammad Alrwaily and hence the corresponding author.
Comment 3: Abstract:
- There is a blank line visible in the abstract. Please connect the paragraphs of the abstract.
- I suggest clearly highlighting the purpose and objectives of the paper in the abstract. Provide an explicit statement of the main objective(s) or question(s) the manuscript addresses.
- I suggest adding keywords such as MSK Careor Virtual Care or Musculoskeletal Conditions.
Response 3: The paragraph is now connected. I have changed the abstracted to include purpose and specific questions as follows:
The purpose of this perspective is to presents a structured framework for delivering musculoskeletal (MSK) care via telerehabilitation, advocating for a fundamental shift in the mindset of physical therapists. Rather than viewing virtual care as a limited substitute, it is redefined as a clinically valid model that requires deliberate reengineering of traditional assessment and treatment practices. The article addresses three key questions: (1) How can MSK assessment and treatment be effectively delivered in the digital environment? (2) What clinical reasoning pathways can guide patient triage in virtual care? and (3) What value does telerehabilitation offer to both patients and therapists?
I have added Musculoskeletal Care and Virtual Care to the Keywords.
Comment 3:
- INTRODUCTION:
- I suggest clearly highlighting the purpose and objectives of the paper in the introduction. Provide an explicit statement of the main objective(s) or question(s) the manuscript addresses.
Response 3: I have clearly indicated the purpose and the questions the paper aims to address in the introduction section:
The purpose of this paper is to explore the value of telerehabilitation, the evolving MSK patient journey, and the clinical shift required from physical therapists. It addresses three key questions: (1) How can MSK assessment and treatment be effectively adapted to the digital environment? (2) What clinical reasoning pathways can guide patient triage in virtual care? and (3) What value does telerehabilitation offer to both patients and therapists across functional, emotional, and systemic dimensions?
Comment 4: - In the introduction, the sentence: “Care for such conditions is challenging due to limited and often inequitable access to timely rehabilitation services.[3]” – is incorrectly cited – the punctuation mark should be after the reference number.
Response 4: Thank you for the catch. This has been corrected.
Comment 5:
- RESHAPING THE MUSCULOSKELETAL CARE PATHWAY THROUGH TELEREHABILITATION
- The chapter includes the sentence: “…the therapist engages the patient as an active participant the evaluation.” It would be good to find or comment on data regarding how often patients are passive participants – how much the focus is on quickly resolving the problem rather than on the patient’s active involvement in their own treatment process.
Response 5: I have included statement into msk pathway: “…marking a notable departure from traditional models where patients have often played a more passive role. In many conventional settings, the emphasis is on therapist-led interventions aimed at symptom resolution, with limited time or structure for shared decision-making or active self-monitoring. Telerehabilitation, by contrast, requires and facilitates greater patient involvement from the outset, encouraging ownership, fostering self-efficacy, and aligning treatment with the patient's lived environment and functional goals. This shift is especially valuable in chronic MSK care, where sustained engagement and behavioral change are often necessary for long-term success.”
Comment 6:
- THE VALUE PROPOSITION OF MSK TELEREHABILITATION
- I suggest standardizing the table presentations. Table 1 is larger than Table 2.
Response 6: I fixed the table sizes.
Comment 7 - Regarding Table 2 (Telerehabilitation Value Proposition for Therapist): “Functionally, it improves clinical efficiency by reducing no-shows, enabling flexible scheduling, and streamlining documentation and follow-up tasks” – it would be good to explain how telerehabilitation simplifies or optimizes documentation, which therapists and healthcare staff are otherwise burdened with. Perhaps support with a reference or provide a personal suggestion.
Response 7: I have added the statement to say: “Documentation, which is often a time-consuming and fragmented process in traditional care, can be simplified through telehealth platforms that offer built-in templates, auto-populated forms, and real-time note-taking features during sessions [9].”
Comment 8:
- PREPARING FOR MSK TELEREHABILITATION SESSION
- This chapter lists several dimensions that must be supported to enable telerehabilitation. It raises the question of whether this is feasible with patients of different age groups, cultures, and languages. It might be desirable to provide an explanation or the authors’ personal suggestion on how sessions could be conducted in such cases, emphasizing technical support for telerehabilitation in older adults and communication – multilingualism in patients from different countries.
Response 8: I have added a statement to Section 13 that say: “To overcome these disparities, several practical solutions can enhance accessibility. Providing simplified interfaces, multilingual platform options, and step-by-step onboarding guides can help reduce cognitive and technical demands. Short pre-session orientation calls and the involvement of caregivers or family members during setup may support digital readiness for older adults”
Comment 9:
- PREPARING FOR PATIENT ENCOUNTER IN TELEREHABILITATION
- The role of the family during the telerehabilitation process is emphasized. It would be good to clarify whether the family also undergoes some form of education to assist during the telerehabilitation process.
Response 9: I added a statement regarding involvement of family member (or caregiver): “When caregivers are present, therapists often provide brief, real-time guidance to help them support the session—such as adjusting camera angles, assisting with movement, or monitoring for safety. While this involvement is usually informal, it effectively enhances care delivery.”
Comment 10 - It states: “…the therapist must confirm the safety of the physical environment” – how objective is it to rely on the patient’s subjective assessment of the space? It might be necessary to clarify whether it is essential that the therapist conducting telerehabilitation visits the patient’s environment, space, and infrastructure in person for a one-day inspection before starting telerehabilitation.
Response 10: I agree that safety is a foundational concern in telerehabilitation. However, I don’t think that therapists should physically inspect the patient's environment prior to initiating care. In many cases, this is not feasible particularly when the therapist and patient are located in different cities or when the patient resides in rural or underserved areas. One of the core advantages of telerehabilitation is its ability to overcome geographical barriers and deliver care remotely.
We added a statement that says: “When needed, the therapist may ask the patient to use their device’s camera to visually scan the room, ensuring there are no hazards such as loose rugs, obstructed pathways, or poor lighting.”
Comment 11:
- SUBJECTIVE ASSESSMENT IN MSK TELEREHABILITATION
- The section emphasizes: “…the therapist must depend on verbal precision and patient narrative” – this raises the question of objectivity in the telerehabilitation process – can we fully rely on the patient’s words even if this means building mutual trust? How can we ensure the reliability of the LMNOPQRST framework?
Response 11: Thank you for raising this important point. I agree that subjectivity and verbal reliance are inherent features of virtual care, and that they introduce certain limitations. However, I argue that the goal in telerehabilitation is not to eliminate subjectivity, but rather to systematically structure it through frameworks like LMNOPQRST, which guide both the clinician and the patient toward a more complete and clinically useful narrative. In this model, clinical trust is not blind acceptance, but a working alliance built on structured inquiry, patient engagement, and critical reasoning. While telerehabilitation may lack the tactile confirmation of hands-on assessment, it compensates through methodical verbal exploration, repeatable self-reports, and continuity over time. We believe the LMNOPQRST model, when applied skillfully, brings valuable clarity and reproducibility to the virtual subjective exam.
Comment 12:
- INTEGRATING OUTCOME MEASURES IN MSK TELEREHABILITATION
- Please explain in more detail how questionnaires and assessment tools will be sent to patients, how they will complete them, and where they will be stored. What will be the frequency of using such tools given that the basic therapist tools, such as palpation and manual measurements, are absent in the telerehabilitation process – please explain the statement “…repeated use of these measures throughout the episode of care…”
Response 12: Thank you for this insightful comment.
Regarding delivery and completion, in most telerehabilitation platforms, outcome measures can be sent electronically to the patient via secure links, mobile apps, or integrated platform dashboards. Patients can complete these forms using their smartphones, tablets, or computers, and submissions are automatically logged into the therapist’s interface or electronic health record. In cases where the platform does not support automated workflows, therapists may email PDF forms or use external tools such as Google Forms, with results uploaded manually into the clinical record.
Regarding storage, data from completed questionnaires are stored within the platform’s secure, system, typically integrated with the patient’s file. This allows for easy retrieval, progress tracking, and comparison over time. Some platforms allow therapists to generate graphs or trend lines, which can be shared with the patient to enhance transparency and engagement.
Regarding frequency, the statement “repeated use of these measures throughout the episode of care” refers to periodic re-administration of selected tools. For example, brief measures such as the Numeric Pain Rating Scale or Global Rating of Change may be used at every session, while longer tools like the Oswestry Disability Index or the Lower Extremity Functional Scale may be administered at baseline, mid-point, and discharge. The frequency is guided by clinical milestones, anticipated changes in symptoms, or the need to document progress for insurance or referral purposes.
I also incorporated details in Section 7 that says:
“When sent electronically in advance via secure links, patient portals, or mobile apps, these measures allow the therapist to review the results ahead of time, identify red flags, and tailor the subjective assessment accordingly”
And
“The frequency of re-administration varies based on clinical need: brief tools may be used at each session, while longer forms may be used at intake, mid-point, and discharge.”
Comment 13:
- ADMINISTERING PSYCHOSOCIAL TOOLS IN MSK TELEREHABILITATION
- It is stated: “Integrating these tools early in care can improve triage accuracy, personalize treatment plans, and flag the need for multidisciplinary support.” – it would be desirable to explain in more detail how psychosocial tools could be implemented throughout the telerehabilitation process and what that would mean for the patient.
Response 13: We added a statement in Section 8 that says: The administration of such tools should not be limited to the initial session but can be repeated at midpoint or near discharge to track changes in cognitive or emotional barriers, support treatment adjustments, and inform decisions about progression.
Comment 14:
- TRIAGING IN MSK TELEREHABILITATION
- In the subsection Online Assessment with In-Person Follow Up, it is stated: “In such cases, the therapist should communicate the need for hybrid care and coordinate appropriately.” – it would be good to explain how the hybrid model would be conducted – would the patient come to the healthcare facility or would the therapist visit the patient? There are many therapists in the private sector who conduct mobile physiotherapy – Physio to Go. Could this form of therapy be implemented in a hybrid telerehabilitation model?
Response 14: Thank you for this valuable comment. I have incorporated a statement that says:
“Hybrid models may include in-person follow-up at a healthcare facility or through mobile physical therapy, where the therapist visits the patient at home. Both settings ensure that hands-on assessment and treatment can complement virtual care when needed.”
Comment 15:
- ADAPTING THE PHYSICAL EXAMINATION FOR MSK TELEREHABILITATION
- Regarding the sentence: “Research shows that the clarity of instruction and video quality directly influence diagnostic accuracy in remote MSK exams” – which research/studies are referred to? Please add references.
- Regarding the sentence: “Literature shows that many of these adaptations preserve high sensitivity and acceptable specificity, especially when interpreted in the context of a solid clinical hypothesis” – which research/studies are referred to? Please add references.
Response 15: references have been added accordingly and highlighted in the text in red.
Comment 16:
- DELIVERING TREATMENT IN TELEREHABILITATION
- I suggest revising Table 4. Place Purpose/Use in the first column, Clinic-Based Tool in the second, and Adaptation for Telerehabilitation in the third. Explain all abbreviations found in the table and include them in the legend.
- Add references for the explanations of Mobility Deficit and Control Deficit.
- I suggest a visual presentation of the three phases of control deficit interventions (Isolated muscle activation, acquisition phase, movement assimilation) and link it to the following chapter on resources (12. THE RESOURCEFUL THERAPIST).
Response 16: I have rearranged Table 4 as suggested. All legends are placed right below Table 4.
References for mobility and control deficits were added
Visual representation of control deficit was added
Comment 17:
- THE RESOURCEFUL THERAPIST
- I suggest linking this chapter with chapter 11. DELIVERING TREATMENT IN TELEREHABILITATION.
Response 17: Thank you for the suggestion. While Chapter 11 and Chapter 12 are closely related, I intentionally chose to keep "The Resourceful Therapist" as a distinct section to highlight it as a standalone concept. This framing emphasizes the mindset shift required in telerehabilitation—not just a change in treatment logistics, but a redefinition of the therapist’s role in creatively navigating limitations, environments, and resources. Keeping it separate allows this idea to stand out and resonate more strongly with the reader.
Comment 18
- CHALLENGES TO THE ADOPTION OF MSK TELEHEALTH
- This section lists numerous barriers and challenges faced by telerehabilitation. The biggest problem is therapist motivation and the traditional physical therapy curriculum. It would be excellent to add potential solutions and/or reflections on how to overcome such barriers that are key for the active implementation of telerehabilitation. It would be highly valuable to include examples from different countries that already actively implement telerehabilitation.
Response 18: Thank you for your comment, we added the following statements to Sections 13:
“To overcome these barriers, academic institutions should incorporate telerehabilitation modules into entry-level curricula and continuing education, including virtual case simulations, remote communication strategies, and technology onboarding. Moreover, experienced clinicians can serve as digital mentors, modeling confidence and success in virtual care delivery. The United Kingdom, for example, has developed national digital MSK pathways through initiatives like the National Health Service MSK Digital Playbook [44], integrating virtual assessment and remote management into care delivery. These models streamline triage, improve access, and support continuity across services. They serve as practical blueprints for scalable telerehabilitation.”
Comment 19:
- SUMMARY
- Instead of “Summary,” it should be “Conclusion.”
Response 19: Thank you for the suggestion. However, since this is a perspective article, we believe the heading “Summary” is more appropriate than “Conclusion.” The term better reflects the intent of the section to synthesize reflections and insights rather than to present definitive findings. We would prefer to retain “Summary” to align with the structure and tone typical of perspective papers.
Comment 20:
Finally, given the complexity and multilayered nature of the approach considered here, we believe that a SWOT analysis based on selected chapters could serve as a useful framework for discussion and critical evaluation. By including such a structured approach — which has already been used in the analysis of various clinical and educational strategies (e.g., in telemedicine*) — it is possible to further clarify the potentials for the broader application of telerehabilitation in practice. It would be important to emphasize in the conclusion that telerehabilitation does not aim to replace the human being.
- * Caponnetto V, Ornello R, De Matteis E, Papavero SC, Fracasso A, Di Vito G, Lancia L, Ferrara FM, Sacco S. The COVID-19 Pandemic as an Opportunity to Improve Health Care Through a Nurse-Coordinated Multidisciplinary Model in a Headache Specialist Center: The Implementation of a Telemedicine Protocol. Telemed J E Health. 2022 Jul;28(7):1016-1022.
- * Pinto M, Gimigliano F, De Simone S, Costa M, Bianchi AAM, Iolascon G. Post-Acute COVID-19 Rehabilitation Network Proposal: From Intensive to Extensive and Home-Based IT Supported Services. International Journal of Environmental Research and Public Health. 2020; 17(24):9335.
Response 20: Thank you for this insightful suggestion. We agree that incorporating a SWOT analysis would provide a structured lens through which the strengths, limitations, opportunities, and threats of MSK telerehabilitation can be more critically appraised. We have included a SWOT table drawing from selected sections of the manuscript. Now there is a new Section 14:
- SWOT ANALYSIS OF MSK TELEREHABILITATION
Given the multilayered nature of telerehabilitation implementation, a structured analysis can help clarify where the model thrives and where further development is required. Drawing on selected insights from this paper, the following SWOT analysis (Table 5) highlights key strengths, weaknesses, opportunities, and threats related to the delivery of musculoskeletal telerehabilitation. This format not only aids critical evaluation but also supports strategic decision-making for clinicians, educators, and policymakers.
Table 5: SWOT Analysis of MSK Telerehabilitation
|
Strengths |
Weaknesses |
|
· Improved accessibility for patients in rural or underserved areas · Flexibility in scheduling and follow-up · Efficient use of validated outcome and psychosocial tools · Opportunities for enhanced documentation and progress tracking · Promotes patient autonomy and engagement through self-management |
· Lack of tactile/manual assessment tools · Variable digital literacy among patients and clinicians · Dependence on patient-reported information and self-performance · Training gaps in PT curriculum regarding virtual assessment and treatment · Environmental constraints in the patient’s home may limit exam or exercise options |
|
Opportunities |
Threats |
|
· Expansion of hybrid care models including mobile physio and home visits · Integration of wearable tech, AI, and asynchronous monitoring · Cross-border consultations and collaborations in global MSK care · Curriculum reform to prepare future therapists for digital health |
· Regulatory ambiguity and inconsistent reimbursement frameworks · Skepticism among therapists and patients regarding the legitimacy of virtual care · Risk of perceived impersonality or depersonalization of care · Technology failures or privacy concerns undermining trust in virtual platforms |
AI: Artificial Intelligence
MSK: Musculoskeletal
SWOT: Strengths, Weaknesses, Opportunities, Threats
I also added a sentence in the summary to say: It is important to emphasize that telerehabilitation is not intended to replace the role of the human therapist, but rather to extend their reach and enhance care delivery through digital means
Reviewer 3 Report
Comments and Suggestions for Authors
This manuscript offers a comprehensive and well-articulated framework for implementing telerehabilitation in musculoskeletal (MSK) care. It presents a conceptual model that reimagines traditional physical therapy practices for virtual environments, emphasizing clinical reasoning, patient empowerment, and digital adaptability. It is clearly written, well-structured, and supported by a robust body of literature, including systematic reviews and clinical guidelines.
However, the manuscript is not a primary research article, nor does it qualify as a structured or systematic review. It does not present original empirical data, nor does it follow a formal review methodology with defined inclusion criteria, search strategy, or data synthesis. Instead, it should be classified as a perspective or practice framework article, based on informal synthesis of existing evidence and clinical experience.
It synthesises a perspective for intervention, including detailed guidance on preparing for virtual sessions, conducting structured subjective assessments, adapting physical examinations, integrating outcome and psychosocial measures, and delivering treatment using household items and digital tools. The triage model for determining appropriate care pathways (remote, hybrid, or referral) is clinically sound and well justified.
Nonetheless, the manuscript would benefit from revisions. First, the abstract does not follow the structured format typically required by the journal. It lacks clear sections for background, objectives, methods, results, and conclusions, which may hinder clarity and indexing. Second, the manuscript should explicitly acknowledge its conceptual nature and clarify that the drafted framework has not yet been empirically validated. Third, a more detailed discussion of the limitations of remote assessment—particularly regarding diagnostic accuracy and feasibility in low-resource settings—would strengthen the manuscript’s transparency and applicability.
It is valuable and forward-looking contribution to the field of MSK rehabilitation. I recommend the manuscript for publication pending minor revisions to clarify its article type, revise the abstract into a structured format, and expand on its limitations and empirical basis.
Author Response
Comment 1:
This manuscript offers a comprehensive and well-articulated framework for implementing telerehabilitation in musculoskeletal (MSK) care. It presents a conceptual model that reimagines traditional physical therapy practices for virtual environments, emphasizing clinical reasoning, patient empowerment, and digital adaptability. It is clearly written, well-structured, and supported by a robust body of literature, including systematic reviews and clinical guidelines.
Response 1: I thank Reviewer 3 for their thoughtful and constructive feedback. I am especially grateful for the recognition of the manuscript’s clinical relevance, clarity, and contribution to advancing musculoskeletal telerehabilitation.
Comment 2:
However, the manuscript is not a primary research article, nor does it qualify as a structured or systematic review. It does not present original empirical data, nor does it follow a formal review methodology with defined inclusion criteria, search strategy, or data synthesis. Instead, it should be classified as a perspective or practice framework article, based on informal synthesis of existing evidence and clinical experience.
Response 2: I agree with the reviewer’s assessment that the manuscript should be positioned as a perspective/practice framework. In response, we have clarified this in both the manuscript title as follows:
Delivering Musculoskeletal Rehabilitation in the Digital Era: A Perspective on Clinical Strategies for Remote Practice
Comment 3:
It synthesises a perspective for intervention, including detailed guidance on preparing for virtual sessions, conducting structured subjective assessments, adapting physical examinations, integrating outcome and psychosocial measures, and delivering treatment using household items and digital tools. The triage model for determining appropriate care pathways (remote, hybrid, or referral) is clinically sound and well justified.
Response 3: Thank you
Comment 4: Nonetheless, the manuscript would benefit from revisions. First, the abstract does not follow the structured format typically required by the journal. It lacks clear sections for background, objectives, methods, results, and conclusions, which may hinder clarity and indexing. Second, the manuscript should explicitly acknowledge its conceptual nature and clarify that the drafted framework has not yet been empirically validated. Third, a more detailed discussion of the limitations of remote assessment—particularly regarding diagnostic accuracy and feasibility in low-resource settings—would strengthen the manuscript’s transparency and applicability.
Response 4: Thank you for your insightful comments and supportive evaluation of our manuscript. We appreciate your suggestion to revise the abstract into a structured format. However, as this is a perspective article, not an original research or systematic review, the structure of the abstract differs by design. That said, we have ensured the revised abstract clearly articulates the background, scope, guiding questions, and intended contributions of the proposed framework. Additionally, in line with your feedback, we have explicitly stated within the manuscript that this framework is a perspective in the title and in the text of the paragraph as follows:
“The purpose of this perspective is to presents a structured framework for delivering musculoskeletal (MSK) care via telerehabilitation, advocating for a fundamental shift in the mindset of physical therapists. Rather than viewing virtual care as a limited substitute, it is redefined as a clinically valid model that requires deliberate reengineering of traditional assessment and treatment practices. The article addresses three key questions: (1) How can MSK assessment and treatment be effectively delivered to the digital environment? (2) What clinical reasoning pathways can guide patient triage in virtual care? and (3) What value does telerehabilitation offer to both patients and therapists?
Comment 5: It is valuable and forward-looking contribution to the field of MSK rehabilitation. I recommend the manuscript for publication pending minor revisions to clarify its article type, revise the abstract into a structured format, and expand on its limitations and empirical basis.
Response 5: Thank you for your comment. We have added a statement in the text to suggest the limitations of remote rehabilitation as follows:
“However, it is important to recognize that certain manual therapy techniques demand precise, skilled hands-on execution that cannot be fully replicated in a virtual setting. Accordingly, certain clinical cases may warrant a hybrid model of care, combining the convenience and continuity of virtual sessions with periodic in-person visits.”
Reviewer 4 Report
Comments and Suggestions for Authors
- There was a visible comment labeled [ZL1] in the submitted manuscript. Please double-check the manuscript to ensure that no internal editing marks remain before submission.
- The manuscript relied heavily on expert opinion. Please clarify which clinical strategies are supported by empirical evidence and which are based on clinical experience.
- Novel adaptations such as self-administered mobilizations and home-based special tests lacked direct citations. If available, provide supporting references. If not, clearly state that these are unvalidated recommendations based on practice.
- The value-proposition section was overly detailed and partially repetitive. Please consider condensing this section and summarizing key examples in a table or appendix.
- Training needs for therapists were mentioned but not elaborated. Please provide specific suggestions on how therapists can acquire the skills required for effective virtual care.
- Barriers related to regulation and reimbursement were listed without further discussion. Please briefly describe practical steps or current trends that may help mitigate these issues.
- The challenges faced by digitally underserved populations were noted only briefly. Expand this point by suggesting ways to improve accessibility, such as simplified platforms or device access.
- The manuscript did not address long-term monitoring or adherence strategies in telerehabilitation. Please propose follow-up methods or outcome-tracking tools for continuity of care.
- Some tables and lists contained excessive detail. Please streamline the content by focusing on core examples and relocating secondary details to supplementary material if needed.
- The manuscript did not include any figures. Please add figures to visually summarize the proposed telerehabilitation framework?
- Some pages of the manuscript contained large areas of blank space. Please adjust the layout for consistency and professional presentation.
- Table 4 appeared to contain formatting inconsistencies, especially in the last three rows. Please review and correct any alignment or spacing issues.
Author Response
Reviewer 4:
Comment 1: There was a visible comment labeled [ZL1] in the submitted manuscript. Please double-check the manuscript to ensure that no internal editing marks remain before submission.
Response 1: The comment was doubled checked and responded to.
Comment 2 & 3: The manuscript relied heavily on expert opinion. Please clarify which clinical strategies are supported by empirical evidence and which are based on clinical experience.
Novel adaptations such as self-administered mobilizations and home-based special tests lacked direct citations. If available, provide supporting references. If not, clearly state that these are unvalidated recommendations based on practice.
Response 2 & 3: I added statement in Section 9 that says: Note that these adaptations are informed by clinical experience and have yet to be supported by empirical validation.
I also added a statement in Section 10 that says: Based on clinical experience
I also added a statement in Section 11.1 that says: Note that these adaptations are informed by clinical experience and yet to be validated in the literature.
Comment 4: The value-proposition section was overly detailed and partially repetitive. Please consider condensing this section and summarizing key examples in a table or appendix.
Response 4: I appreciate the suggestion to condense the value-proposition section. However, based on earlier feedback from Reviewers 1 and 2, I expanded this section to provide more detail to accommodate their requests. I believe this level of detail supports the practical utility of the framework and highlights the distinct benefits for both patients and therapists. Nonetheless, we have carefully reviewed the section again to minimize redundancy and improve clarity, while preserving the illustrative content that prior reviewers found valuable.
Comment 5: Training needs for therapists were mentioned but not elaborated. Please provide specific suggestions on how therapists can acquire the skills required for effective virtual care.
Response 5: I added a statement in Section 13 that says: “To overcome these barriers, academic institutions should incorporate telerehabilitation modules into entry-level curricula and continuing education, including virtual case simulations, remote communication strategies, and technology onboarding. Moreover, experienced clinicians can serve as digital mentors, modeling confidence and success in virtual care delivery.”
Comment 6: Barriers related to regulation and reimbursement were listed without further discussion. Please briefly describe practical steps or current trends that may help mitigate these issues.
Response 6: I added a statement in Section 13 that says: “However, current trends indicate gradual progress. Several countries have extended temporary telehealth provisions introduced during the pandemic into permanent policy. For example, Australia and parts of the United States have introduced telehealth parity laws and unified billing codes to ensure virtual care is reimbursed comparably to in-person care. Moreover, professional associations such as the American Physical Therapy Association have issued guidance to support therapists in navigating evolving regulations and documentation practices.”
Comment 7: The challenges faced by digitally underserved populations were noted only briefly. Expand this point by suggesting ways to improve accessibility, such as simplified platforms or device access.
Response 7: In Section 13, I added a statement to address this: “To overcome these disparities, several practical solutions can enhance accessibility. Providing simplified interfaces, multilingual platform options, and step-by-step onboarding guides can help reduce cognitive and technical demands. Short pre-session orientation calls and the involvement of caregivers or family members during setup may support digital readiness for older adults. Countries such as Australia, the United Kingdom, and Canada offer scalable models through public investment, national digital MSK pathways, and integrated rural strategies that serve as practical blueprints for inclusive telerehabilitation.”
Comment 8: The manuscript did not address long-term monitoring or adherence strategies in telerehabilitation. Please propose follow-up methods or outcome-tracking tools for continuity of care.
Response 8: I added a statement under Section 11.3 to say: 11.3. Continuity of Care and Long-Term Monitoring
To support long-term monitoring and adherence in telerehabilitation, several strategies can be integrated into the digital care model. These include the use of outcome-tracking tools such as patient portals and mobile applications, which allow individuals to regularly log symptoms, pain levels, and activity adherence. Wearable devices can provide objective data on movement, activity, and biometrics, which can be shared with clinicians for ongoing assessment. Automated reminders and digital alerts help maintain exercise compliance, while asynchronous check-ins such as periodic surveys or video updates ensure continued engagement.
Data dashboards within telerehabilitation platforms can visualize patient progress, flag setbacks, and support adaptive care planning. Additionally, gamification strategies, including progress milestones and motivational prompts, may enhance patient engagement and motivation over time. Collectively, these approaches allow telerehabilitation to deliver continuity of care beyond scheduled sessions and support sustainable self-management.
Comment 9: Some tables and lists contained excessive detail. Please streamline the content by focusing on core examples and relocating secondary details to supplementary material if needed.
Response 9: Thank you for the observation. I recognize the importance of clarity and conciseness in presenting tables and lists. However, some degree of detail was intentionally retained based on feedback from other reviewers. If you could kindly specify which tables or lists you feel contain excessive detail, I would be glad to revise or move content to supplementary material accordingly.
Comment 10: The manuscript did not include any figures. Please add figures to visually summarize the proposed telerehabilitation framework?
Response 10: I have created a figure that summarizes the MSK telerehabilitation framework. I also referred to the figure earlier in the text under Section 4
Comment 11: Some pages of the manuscript contained large areas of blank space. Please adjust the layout for consistency and professional presentation.
Response: 11: all blank spaces are adjusted. Please note that sometimes due to size of table or figure it would be hard to eliminate the white space.
Comment 12: Table 4 appeared to contain formatting inconsistencies, especially in the last three rows. Please review and correct any alignment or spacing issues.
Response 12: Table 4 corrected and adjusted.
Round 2
Reviewer 1 Report
Comments and Suggestions for Authors
The references still do not comply with the journal's guidelines.
Author Response
Reviewer 1:
Comment: The references still do not comply with the journal's guidelines.
Response: Thank you for your note. The references have now been updated to match the journal’s guidelines. Originally, some journal titles were left in full because, according to the journal’s instructions, “If you are not sure how to abbreviate a particular journal title, please leave the entire title.” In this revision, journal titles have been manually abbreviated according to ISO 4 rules.
Reviewer 4 Report
Comments and Suggestions for Authors
The author has addressed my concern and the manuscript can be accepted.
Author Response
Reviewer 4:
Comment: The author has addressed my concern and the manuscript can be accepted.
Response: Thank you very much!